# Modeling intensive ocean-cryosphere interactions in Lützow-Holm Bay, East Antarctica

Kazuya Kusahara[1], Daisuke Hirano[2, 3], Masakazu Fujii[4, 5], Alexander D. Fraser[6], Takeshi Tamura[4, 5]

[1]Japan Agency for Marine-Earth Science and Technology (JAMSTEC), Yokohama, Kanagawa, 236-0001, Japan
5    [2]Institute of Low Temperature Science, Hokkaido University, Sapporo, Hokkaido, 060-0819, Japan
[3]Arctic Research Center, Hokkaido University, Sapporo, Hokkaido, 001-0021, Japan
[4]National Institute of Polar Research, Tachikawa, Tokyo, 190-8518, Japan
[5]Graduate University for Advanced Studies (SOKENDAI), Tachikawa, Tokyo, 190-8518, Japan
[6]Australian Antarctic Program Partnership, University of Tasmania, Hobart, Tasmania, 7004, Australia

*Correspondence to*: Kazuya Kusahara (kazuya.kusahara@gmail.com, kazuya.kusahara@jamstec.go.jp)

List of ORCID

Kazuya Kusahara          : 0000-0003-4067-7959

15   Daisuke Hirano          : 0000-0002-8047-1544

Masakazu Fujii          : 0000-0003-0527-1742

Alexander D. Fraser     : 0000-0003-1924-0015

Takeshi Tamura          : 0000-0001-8383-8295

**Abstract.** Basal melting of Antarctic ice shelves accounts for more than half of the mass loss from the Antarctic Ice Sheet. Many studies have focused on active basal melting at ice shelves in the Amundsen-Bellingshausen Seas and the Totten Ice shelf, East Antarctica. In these regions, the intrusion of Circumpolar Deep Water (CDW) onto the continental shelf is a key component for the localized intensive basal melting. Both regions have a common oceanographic feature: southward deflection of the Antarctic Circumpolar Current brings CDW toward the continental shelves. The physical setting of Shirase Glacier Tongue (SGT) in Lützow-Holm Bay corresponds to a similar configuration in the southeastern side of the Weddell Gyre in the Atlantic sector. Here, we conduct a 2–3 km resolution simulation of an ocean-sea ice-ice shelf model using a recently-compiled new bottom topography dataset in the bay. The model can reproduce the observed CDW intrusion along the deep trough. The modeled SGT basal melting reaches a peak in summer and a minimum in autumn and winter, consistent with the wind-driven seasonality of the CDW thickness in the bay. The model results suggest the existence of eastward-flowing undercurrent on the upper continental slope in summer, and the undercurrent contributes to the seasonal-to-interannual variability of the warm water intrusion into the bay. Furthermore, numerical experiments with and without fast-ice cover in the bay demonstrate that fast ice plays a role as an effective thermal insulator and reduces local sea-ice formation, resulting in much warmer water intrusion into the SGT cavity.

## 1 Introduction

The Antarctic ice sheet is the greatest freshwater reservoir on the present-day's Earth surface, most of which sits upon bedrock. The exchanges of the mass, heat, and freshwater between the Antarctic ice sheet and the ocean have played a critical role in regulating the global sea level and influencing the Earth's climate (Convey et al., 2009; Turner et al., 2009). Recent satellite observations have revealed a declining trend in the Antarctic ice sheet mass (Paolo et al., 2015; Rignot et al., 2011, 2019). Several observational and modeling studies have pointed out that enhanced intrusions of warm waters onto some Antarctic continental shelf regions trigger more active ocean-ice shelf interaction (i.e., ice-shelf basal melting), leading to the negative mass balance of the Antarctic ice sheet (Depoorter et al., 2013; Rignot et al., 2013).

Circumpolar Deep Water (CDW) is the warmest water mass at intermediate depths at all longitudes over the Southern Ocean, and the intrusions onto the Antarctic continental shelf regions result in the very active basal melting at the Antarctic ice shelves. Jacobs et al. (1996), in the first in-situ oceanographic observation in front of the Pine Island Glacier, reported that an intrusion of the CDW onto the Amundsen continental shelf provides large heat flux to melt the ice shelf base. After that time, many studies have shown evidence for active basal melting at the ice shelves in the Pacific sector (Jenkins et al., 2018), and the ice shelves in this sector have been relatively well studied, along with the local bottom topography and the surrounding oceanic conditions.

More recently, several studies have started to focus on the Totten Ice shelf, Wilkes Land, East Antarctica, as a newly-recognized hot spot of active ice shelf-ocean interaction caused by a CDW intrusion (Rintoul et al., 2016; Silvano et al., 2016, 2018, 2019). Ice shelves in the Amundsen and Bellingshausen Seas and the Totten ice shelves have a common feature: the Antarctic Circumpolar Current (ACC) is proximal to the Antarctic continental shelf regions, and thus the CDW can affect regional coastal water masses. The Southern Ocean has three large-scale cyclonic (subpolar) gyres: Ross, Kerguelen, and Weddell Gyres (Gordon, 2008). Amundsen and Bellingshausen Seas and the offshore region of Wilkes Land correspond to south-eastern sides of the Ross and Kerguelen Gyres, respectively, where the southward deflection of the large-scale ocean circulations brings warm water poleward to the Antarctic continental shelves (Armitage et al., 2018; Dotto et al., 2018; Gille et al., 2016; McCartney and Donohue, 2007; Mizobata et al., 2020). By analogy to the relationship between the southward deflection of the ocean circulations and the active ice shelf melting areas, one can naturally speculate active ocean-ice shelf interaction around the south-eastern side of the Weddell Gyre (Ryan et al., 2016). In fact, looking at an estimate of basal melt rate from Rignot et al. (2013), a glacier tongue (Shirase Glacier Tongue, SGT) exhibits a high basal melt rate of $7\pm2$ m yr$^{-1}$, although the areal extent is small. The upstream glacier, Shirase Glacier in Lützow-Holm Bay (LHB), Enderby Land, East Antarctica, is one of the fastest laterally-flowing glaciers around Antarctica (Nakamura et al., 2010; Rignot, 2002).

The coastal Japanese Antarctic station, Syowa Station, is located within LHB, and it is the main platform of the Japanese Antarctic Research Expedition (JARE) for Japanese Antarctic sciences. LHB is bounded to the east by the north-south oriented coastline and to the west by northwest-southeast coastline/ice shelf (Fig. 1), and the semi-closed bay is open to the north. As an interesting feature, LHB is usually covered with multi-year fast ice throughout the year (Fraser et al., 2012, 2020; Nihashi and Ohshima, 2015; Ushio, 2006). The fast-ice cover has made in-situ oceanographic observations difficult. Notably, JARE (31st and 32nd expeditions from 1990 to 1992) conducted year-round oceanographic observations through drilled holes on the fast ice in LHB. Ohshima et al. (1996) analyzed these observation data and investigated the seasonal cycle of Winter Water (WW), which is surface-to-subsurface cold water formed in a winter mixed layer. Furthermore, they proposed a physical mechanism of seasonal change in the WW thickness: the strength of the alongshore wind stress controls the WW thickness through Ekman convergence. At that time, although the study did not focus on warm water near the

bottom, their ocean observations in the bay clearly show the presence of CDW (characterized by high temperatures > 0.0°C, and high salinity > 34.5 psu) below the WW (characterized by the near surface-freezing temperatures). Stronger winds in autumn and winter increase Ekman convergence (downwelling), thickening the WW in the surface to subsurface layers and thinning a warm water layer below the WW. Conversely, weaker winds in the summer season relax the Ekman convergence, leading to a relatively thin WW layer and a thicker CDW layer in deep layers.

Although multi-year fast ice usually covers the southern part of LHB, the fast ice has experienced extensive breakup events with irregular intervals of a few decades (Aoki, 2017; Ushio, 2006). A relatively large breakup occurred from March to April 2016. As a result of the breakup event, 58th JARE could perform an extensive campaign of trans-bay ship-based hydrographic observations to investigate possible ocean-ice shelf/glacier interaction in LHB (Hirano et al., 2020). The in-situ oceanographic observations in the bay elucidate intrusion of CDW onto the continental shelf, and the warm-water signal extends to the front of the SGT, indicating active ice-ocean interaction at SGT.

As mentioned above, LHB and SGT appear to be a suitable place for examining Antarctic ocean-ice shelf interaction in the Atlantic sector. However, they have been overlooked until very recently. Although the recent oceanographic observations show the existence of CDW in the bay (Hirano et al., 2020), little is known about the variability of the warm water intrusions, and it is not clear how the intrusion is controlled. As far as we know, there is no numerical modeling study that focuses on the water mass formation/exchange and the ocean-ice shelf interaction in this region. In this study, we perform a numerical simulation to illustrate coastal water masses and ocean-ice shelf interaction in LHB. In particular, we focus on the seasonal-interannual variability of warm water intrusion from shelf break to the bay and basal melting at SGT. Additionally, we investigate the roles of fast ice on ocean conditions and the ice-ocean interaction in this area.

## 2 Numerical model, bottom topography, and atmospheric conditions

### 2.1 An ocean-sea ice-ice shelf model for Lützow-Holm Bay

This study utilized a coupled ocean–sea ice–ice shelf model (Kusahara and Hasumi, 2013, 2014). The model employed an orthogonal, curvilinear, horizontal coordinate system. Two singular points of the horizontal curvilinear coordinate in the model were placed on the East Antarctic continent (72°S, 30°E and 69°S, 50°E) to regionally enhance the horizontal resolution around the LHB region while keeping the model domain circumpolar Southern Ocean. An artificial northern boundary was placed at approximately 30°S. The same technique to enhance the regional horizontal resolution has been used for several studies on Antarctic coastal ocean modeling (Kusahara et al., 2010, 2017; Marsland et al., 2004, 2007).

The vertical coordinate system of the ocean model was z coordinate. There are four vertical levels of 5-m cell thickness below the ocean surface, and 49 levels of 20-m thickness in the depth range from 20 m to 1000 m. Below this depth, the vertical spacing varies with depth (50-m thickness for 1000–2000 m, 100-m thickness for 2000–3000 m, and 200-m thickness for 3000–5000 m). The maximum ocean depth in the model was set to 5000 m to save computational resources. The model has an ice shelf component for the z-coordinate system (Losch, 2008). A partial step representation was adopted for both the bottom topography and ice shelf draft to represent them optimally in the z-coordinate ocean model (Adcroft et al., 2002). The sea-ice component used one-layer thermodynamics (Bitz and Lipscomb, 1999) and a two-category ice thickness representation (Hibler, 1979). Prognostic equations for the sea-ice momentum, mass, and concentration (ice-covered area) were taken from Mellor and Kantha (1989). Internal ice stress was formulated by the elastic-viscous-plastic rheology (Hunke and Dukowicz, 1997), and sea ice salinity was fixed at 5 psu. Unstable stratification produced in the ocean

model was removed by a convective adjustment, and a surface mixed layer scheme (Noh and Kim, 1999) was used for the open ocean, as in our previous study of a circumpolar Southern Ocean modeling that showed a reasonable representation of seasonal mixed layer depth over the Southern Ocean (Kusahara et al., 2017).

In the ice shelf component, we assumed a steady shape in the horizontal and vertical directions. The freshwater flux at the base of ice shelves was calculated with a three equation scheme, based on a pressure-dependent freezing point equation and conservation equations for heat and salinity (Hellmer and Olbers, 1989; Holland and Jenkins, 1999). We used the velocity-independent coefficients for the thermal and salinity exchange velocities (i.e., $\gamma_t = 1.0 \times 10^{-4}$ m s$^{-1}$ and $\gamma_s = 5.05 \times 10^{-7}$ m s$^{-1}$, Hellmer and Olbers (1989)), although ocean velocity-dependent coefficients have been often used in the recent ice shelf-ocean modeling studies. Velocity magnitudes under the ice shelves strongly depend on the horizontal and vertical grid resolutions (Gwyther et al., 2020), and thus we preferred using the velocity-independent version to minimize the dependency originated from the model configuration. The modeled meltwater flux and the associated heat flux were imposed on the ice shelf-ocean interface.

The horizontal grid spacing over the LHB region was less than 2.5 km (Fig. 1). This relatively high horizontal resolution enabled us to accurately reproduce coastline, ice front line, and bottom topography in the focal region. The background bathymetry for the Southern Ocean in this model was derived from the ETOPO1 (Amante and Eakins, 2009), while the ice shelf draft and bathymetry under the ice shelf were obtained from the RTopo-2 dataset (Schaffer et al., 2016). The bottom topography in the LHB bay region (35–40°E, and 70–68°S) was replaced with a detailed topography that blended multi-beam surveys from the 51st–55th JARE expeditions, point echo sounding using sea ice drill holes from the 9th–22nd JARE expeditions (Moriwaki and Yoshida, 1983), depth data from the nautical chart created by Japan Coast Guard (JCG), in which ship-based single-beam echo sounding data obtained in several JARE expeditions were included, and ETOPO1 (Amante and Eakins, 2009). The detail for the compiled bottom topography was described in the Method section in Hirano et al. (2020). In a later subsection 2.2, we compare this recently-compiled bottom topography with several topography datasets.

As mentioned in the introduction, extensive fast ice has been identified along the East Antarctic coast (Fraser et al., 2012; Nihashi and Ohshima, 2015), and the existence of persistent fast ice characterizes LHB. We introduced areas of multi-year fast ice cover into LHB in the model as constant-thickness (5 m) ice shelf grid cells. Although, in reality, the horizontal distribution and thickness of fast ice vary seasonally and interannually (Aoki, 2017; Fraser et al., 2012; Ushio, 2006), the spatial distribution of multi-year fast ice in the model was assumed to be constant in time as a first approximation. This assumption is likely more valid here in LHB than in more dynamic areas of fast ice cover, where breakouts can occur many times per year. The fast ice in the model prohibited momentum, heat, and freshwater fluxes at the ocean surface with the atmosphere. Since the fast ice in the model was treated as thin ice-shelf, the fast ice also provided freshwater to the ocean surface, based on the melt rate diagnosed with thermodynamic fast ice-ocean interaction.

The model's initial conditions of temperature and salinity were derived from the Polar Science Center Hydrographic Climatology (Steele et al., 2001), with zero ocean velocity over the model domain. Note that the water properties in the climatology over the Southern Ocean come from World Ocean Atlas 1998 (Conkright et al., 1999). North of 40°S, ocean temperature and salinity were restored to the monthly mean climatology throughout the water column with a damping time scale of 10 days. Outside of the focal region where the horizontal resolution becomes coarser than 10 km, sea surface salinity was restored to the monthly mean climatology to suppress unrealistic deep convection in some regions (e.g., the Weddell Sea).

Daily surface boundary conditions for the model were surface winds, air temperature, specific humidity, downward shortwave radiation, downward longwave radiation, and freshwater flux. To calculate the wind stress and sensible and latent heat fluxes, we used the bulk formula of Kara et al. (2000). When the surface air temperature was below 0°C, precipitation was treated as snow. Daily reanalysis atmospheric conditions were calculated from the ERA-Interim dataset (Dee et al., 2011). The ocean-sea ice-ice shelf model was first integrated for 20 years with the 2005-year forcing for spin-up to obtain a quasi-steady state in the ocean and cryosphere components.

We performed three baseline experiments: "CKDRF", "FI", and "NOFI" cases. All three experiments started from the final state of the spin-up integration. The effect of fast ice was included in the first two experiments (CKDRF and FI cases), but not in the NOFI case. In the CKDRF case, the model was continued to be driven by the 2005-year surface forcing to check a model drift in the experiments. In the FI and NOFI cases, a hindcast simulation was carried out for the period 2006–2018 with interannually varying forcing. The difference between the FI and NOFI cases allows us to examine the role of fast ice on the ocean and cryosphere components. It should be noted that the two icescape configurations were largely simplified compared to the actual time-varying fast ice distribution. The numerical results from the two extreme cases were utilized to examine the transient response of the ocean and cryosphere components from the FI case to the NOFI case and the impact of the fast ice on the quasi-equilibrium state.

In the CKDRF and FI case, we used the same coefficients of the ice-shelf thermal and salinity exchange velocities, even for the fast ice-ocean interaction. The coefficients of the exchange velocities were based on observational facts (Hellmer and Olbers, 1989), and the magnitude corresponds to ocean velocity of 15–20 cm/s under ice shelves in velocity-dependent parameterization (Holland and Jenkins, 1999). Under the ice shelf, the velocity-independent parametrization with the coefficients is partially justified with the strong tidal flows as an unresolved process in the models without tidal forcing. However, since fast ice is located in a more open area, this parameterization is expected to overestimate the basal melting at fast ice. To investigate the sensitivities of the fast ice melting and the impact on ice-shelf melting and environmental ocean conditions with respect to the exchange coefficients, we conducted three additional sensitivity experiments with multiplying factors of 0.5, 0.2, and 0.0 on the thermal and salinity exchange coefficients (FIMOD05, FIMOD02, and FIMOD00) only under the fast ice region. In the FIMOD00 case, although there are no heat, salinity, freshwater exchanges between fast ice and ocean, ice shelves (e.g., SGT) can melt as in the baseline experiments.

## 2.2 Bottom topography in Lützow-Holm Bay

The accuracy of bottom topography is critically vital for high-resolution ocean modeling because the geostrophic contour/bottom topography strongly controls ocean flows (including CDW intrusions on the East Antarctic continental shelf, e.g., Nitsche et al. (2017)) and the associated water properties. Several large-scale compilations of bottom topography are now available. As an example, Figure 2b–f shows spatial distributions of bottom topography in our focal region in five bottom topography datasets: ETOPO1 (Amante and Eakins, 2009), ETOPO5 (NOAA, 1988), GEBCO (IOC, IHO and BODC, 2003), RTopo-1 (Timmermann et al., 2010), and RTopo-2 (Schaffer et al., 2016). As can be seen in Fig 2, there are considerable differences in the representations of bottom topography, in particular in the southern half of LHB where fast ice is particularly present. All of the products have a sill-like feature at around 37.0°E and 68.4°S and depression-like features south of it. In the five products, ETOPO1 captures the large-scale features when comparing with JARE's in-situ depth observations (see dots in Fig 2a). For this reason, the ETOPO1 dataset was utilized for the background topography for the recently-compiled topography (Hirano et al., 2020).

Looking at the recently-compiled topography, there are two submarine canyons on both sides of the sill at the latitudes of 68.4°S. It should be noted that the bathymetric features around the sill, including the submarine canyons, are well constrained by the observed depths derived from the point echo sounding and the hydrographic chart (colored dots in Fig. 2a). Acoustically observed data confirms that the sill top is ~500 m deep and surrounded by at least ~850 m deep western submarine canyon and 1600 m deep eastern submarine canyon. South of the sills, there is a large depression whose maximum depth is deeper than 1000 m. From the depression, four troughs are extended southward. The most eastern trough (labeled with T4) is the longest and deepest among them and extends toward SGT. In the southern part of the trough, there is a north-southward elongated depression at the latitudes from 69.5°S to 70°S with a maximum depth of approximately 1600 m. The feature is confirmed in the observational data. We consider that our dataset is the best estimate of bottom topography in the LHB region at this moment, in which available bathymetric observations were taken into account.

## 2.3 Atmospheric conditions

In this subsection, we show seasonal changes in the long-term average fields of mean sea level pressure (MSLP) over the Southern Ocean and surface winds off the LHB region (Fig. 3), based on the ERA-Interim climatology for our main analysis period 2008−2018. A circumpolar pattern of low pressure at high southern latitudes and high pressure at lower latitudes accompanies a prevailing westerly wind over the Southern Ocean in the latitudes from 40°S to 60°S, being a predominant spatial pattern in the high-latitudes Southern Hemisphere. There are three regional minimums of the MSLP in the Weddell Sea, the Indian Sector, and the Amundsen Sea, and the regional minimums deepen in the autumn and winter seasons. South of the air pressure minimums (i.e., Antarctic coastal regions) is a prevailing easterly wind regime, which is the driving force of westward flowing currents along the Antarctic coastal margins (Thompson et al., 2018).

LHB is located on the south-eastern side of the low pressure in the Weddell Sea (see the red box in Fig. 3). In the longitude bands, the westerly wind blows north of 62°S, and the easterly wind blows south of 64°S throughout a year (Fig. 3e–h). The alongshore component dominates the easterly wind, and the wind speed reaches the maximum in autumn (from March to June) and the minimum in the spring and summer seasons (from October to January).

## 3 Difference in the surface forcing in the experiments with and without fast ice cover

Fast ice is formed by sea ice fastening to the Antarctic coastline and/or ice-shelf front, and the typical thickness is a few meters (Fraser et al., 2012; Giles et al., 2008). Note again that the fast ice in this model is represented as an uppermost-one-grid ice shelf. Fast ice becomes a physical barrier between ocean and atmosphere, modifying the exchanges of surface heat, freshwater, and momentum fluxes. The southern half of LHB is usually covered with fast ice, except during the breakup events. Before examining the effects of the fast ice cover on the ocean conditions in LHB and the SGT melting in the next section, we show the difference in the surface forcing with and without fast ice cover.

Sea ice production is a key proxy for cold and saline water mass formation along the Antarctic coastal margins (Morales Maqueda et al., 2004; Tamura et al., 2008, 2016) because brine rejection (very-high-salinity water from sea ice to the ocean when sea ice is formed) is proportional to the sea ice production. The newly-formed cold and saline water mass destratifies the local water columns over the Antarctic coastal regions. Perennial fast ice is a very effective thermal insulator between the atmosphere and ocean due to its thickness (N.B. a perfect insulator in the model, so no atmosphere/sea ice-ocean interaction). Therefore, we investigate the difference in sea ice production in the FI and NOFI cases (Fig. 4). Of course, there is zero sea

ice formation under the fast ice in the LHB region in the FI case (Fig. 4a). In the NOFI case, relatively high sea ice production areas are produced along the eastern coastline in the bay (Fig. 4b). The sea ice production in the NOFI case reaches a maximum in autumn (April–May), and the active sea ice production areas are confined in the coastal regions within 30 km distance from the coastline or the ice front (Fig. 5a). The difference in sea ice production in the fast ice region plays a major role in regulating the vertical profile of the coastal water masses and the subsequent SGT basal melting, as shown later.

Ocean melts the fast ice base as well as ice shelves, and thus the fast ice basal melting provides freshwater to the ocean surface. The basal melting depends on the magnitudes of the thermal and salinity exchange coefficients at the fast ice-ocean interface. Basal melt rate and amount of the fast ice and ice shelves/glaciers in the experiments are summarized in Table 1. Here, we briefly comment on the feasibility of the modeled basal melt rate at the fast ice and examine the freshwater (meltwater) forcing under the fast ice in the experiments. The climatological average precipitation is about 60 cm water equivalent per year (2008–2018 average in the ERA-Interim dataset). This surface input is used here to gauge the feasibility of the estimated magnitude of the fast ice basal melting in the model. The mean melt rate of the fast ice in the FI case is estimated to be 2.29 m $yr^{-1}$, which can not be balanced by the local precipitation. With decreasing the magnitude of the exchange velocities in the FIMOD05 and FIMOD02 cases, the fast ice's mean melt rates are declined to 1.50 m $yr^{-1}$ and 0.83 m $yr^{-1}$, respectively. Taking account of the inputs from the dynamical interaction with drifting sea ice (Fraser et al., 2012; Massom et al., 2010) and the snow-ice formation of the fast ice (Zhao et al., 2020), the melt rate in the FIMOD02 seems to be in a realistic range. The modeled ocean velocities just under the fast ice vary from a few cm $s^{-1}$ to 15 cm $s^{-1}$ depending on the location and season, with the approximate mean speed of 5 cm $s^{-1}$. The exchange coefficients in the FIMOD02 case correspond to the relative ice-ocean velocity of 3–4 cm $s^{-1}$, which is roughly consistent with the modeled ocean flow speed under the fast ice.

In this study, we use results from the FI case to represent the baseline ocean-cryosphere conditions in the existence of the fast ice, taking account of the continuity from the spin-up integration, although the basal melting at the fast ice is exaggerated (Table 1). In the several following analyses and figures, we show the results from the FIMOD series as well as the baseline experiments to show the dependency of the modeled ocean-cryosphere conditions on the different fast ice melting. As a result (shown later), the impacts of the fast ice meltwater on the ocean conditions and the SGT basal melting are much smaller than that of sea ice production difference in the experiments with and without the fast ice cover.

**4 Warm water intrusion into LHB and active basal melting at SGT**

Since the model started from a motionless state with the climatological ocean properties, it requires a spin-up time to adjust simulated ocean circulation and water masses in LHB to a quasi-steady state. In this study, we can use the time series of the basal melt rate at SGT to monitor the spin-up of the ocean-cryosphere system in LHB (Fig. 6) because the basal melting is an integrated result from the interaction between the ocean flow and coastal water masses. As can be seen in Fig. 6a, the basal melt rate reaches a quasi-steady state after approximately 20-year. There is a slightly declining trend in the basal melt rate after the 20-year spin-up, due to an inherent model drift or insufficient spin-up for the circumpolar Southern Ocean. However, the magnitude of the declining trend of the melt rate in the CKDRF case is much smaller than interannual variability and the difference between the FI and NOFI cases (Figs. 6b and 7). Therefore, we can use the results from the two baseline experiments to examine the seasonal and interannual variability.

## 4.1 Transient response and quasi-equilibrium of the CDW intrusion into LHB and the SGT melting

The NOFI case corresponds to the sudden removal of all the fast ice in LHB on January 1, 2006. The main difference of the model configuration between the FI and NOFI cases is the surface boundary conditions in the fast ice-covered area, as explained in the previous section. In this section, firstly, we compare the modeled ocean properties in both the FI and NOFI cases with the recent hydrographic observations in LHB (Hirano et al., 2020). The in-situ observations were conducted in 2017 January, and it means 8–9 months after the extensive fast ice breakup in austral autumn (March–April) of 2016 (Aoki, 2017). The observations seem to be done in a transient phase from the large breakup event. Using the time evolution of the differences in the ocean properties and the SGT melting between the FI and NOFI cases, we roughly estimate a transient timescale of the ocean-cryosphere system responding to the sudden fast ice removal. After showing the transient timescale of a few years, finally, we use monthly climatology for the ocean and cryosphere components over the period 2008–2018 to examine the difference in the quasi-equilibrium states in the FI and NOFI cases.

Figure 8 shows key sections and the observational stations used in this study, with the spatial distributions of the modeled bottom temperature in January and the annual mean basal melt rate at the ice shelves/glaciers. The TROUGH section is a north-southward running observational line from the deep depression to the SGT ice front, and the A-line is located at approximately 69.58°S. Warm water intrusions into LHB from the shelf break are clearly seen in Fig. 8 (for the NOFI case), and the warmest waters are identified along the four troughs (T1, T2, T3, and T4 from west to east). Warm water in the trough T4 (e.g., along the TROUGH section) extends southward toward SGT and Skallen Glacier (SkG).

In both the FI and NOFI cases, the vertical profiles of the ocean properties along the TROUGH section are a clear two-layer structure consisting of a cold-fresh surface layer and a warm-saline deep layer (Fig. 9). Colored circles in Fig. 9 are subsampled ocean properties from the in-situ observations (Hirano et al., 2020). The two-layer feature in the model is consistent with the observation results. In particular, the model can reasonably represent the inflowing warm water temperature in the bottom layer along the trough (Figs. 9a and 9b). The temperature of the warm water intrusion is much warmer than the local freezing point and thus a driver of high basal melting at SGT. The warm water intrusion in the FI case (Fig. 9a) is thicker than in the NOFI case (Fig. 9b), and the temperature difference between the two cases becomes the largest in a subsurface-intermediate depth range (50–400 m). The observed ocean temperature in the subsurface-intermediate depths is colder than −1.0°C, indicating that the temperature profile in the observation is more similar to that in the NOFI case. The existence of the relatively cold water in the upper layers suggests that the water in LHB has experienced surface cooling after the fast ice breakup. Looking at the salinity profiles, again, the model represents the clear intrusion of high salinity signal along the bottom from the sill to the SGT (Fig. 9c and 9d). There are some model biases in salinity. The observed salinity near the bottom throughout the section is higher by more than 0.1 psu than in the model. Since the less saline signal in the model is found near the sill region, the salinity biases probably come from an insufficient representation of salinity values on the continental slope and rise regions. Even taking into account the salinity bias throughout the section, the near-bottom salinity at the southernmost stations (E1 and A3) is substantially underestimated in both cases, in particular in the NOFI case. The near-bottom salinity in the FI case is higher than in the NOFI case, and thus the FI case appears to better represent the near-bottom water salinity at the southernmost stations. The better agreement of the bottom salinity between the observation and FI case suggests that the near bottom signal responds more slowly to the fast ice breakup and that the in-situ observations within one year from the breakup event captured the transitions of the coastal water masses.

Next, we try to estimate a transition timescale of the ocean-cryosphere system in LHB from the breakup to fast ice-free condition, using time evolution of deviations of the model results in the NOFI case from those in the FI case (NOFI − FI). Figure 10 shows the time evolution of the ocean temperature difference at stations G3, E4, and A4 on the TROUGH section

(Figs. 8 and 9) and the SGT basal melt rate. Just after removing the fast ice (i.e., 2006 in the NOFI case), the surface layers are warmed up (clearly seen at the stations E4 and A3), exposing the ocean surface to a warm atmosphere and downward shortwave radiation in summer. In the subsequent winter, the ocean experiences a substantial cooling from the surface to intermediate depths due to sea ice formation in the bay (Fig. 4b). The difference in the temperature profile appears to be stable in a few years after removing the fast ice. This is also confirmed in the difference of the SGT basal melt rate (Fig. 10d). The basal melt rate is rapidly decreased by approximately 4 m yr$^{-1}$ in the first winter, and then it continues to decline in a few years. Comparing with the melt rate difference in the last six years when the equilibriums are assumed to be reached in the FI and NOFI cases (black line and grey shade in Fig. 10d showing the mean and the standard deviations of the melt rate difference between the two cases), the melt rate deviation is in the standard deviation range after a few years.

In the end of this subsection, we compare the ocean temperature profile along the A-line between the observation and NOFI case (Fig. 11). As shown above, the observations seem to capture the transient conditions. Since the observed temperature profile along the TROUGH section is more similar to that in the NOFI case (Fig. 9), here we use the results from the NOFI case for the comparison along the A-line. The vertical section of potential temperature along the A-line with the normal velocity (positive for northward) illustrates that the warm water intrusion is strongly trapped near the bottom on the eastern flank of the deep trough (Fig. 11). In January, the warm water ($> 0°C$) is present in the bottom layer denser than 27.6 kg m$^{-3}$, and the magnitude of the southward flow is larger than 10 cm s$^{-1}$ (Fig. 11a). On the western side of the trough, the northward flow of cold and fresh water is reproduced at the surface and intermediate depths. The overturning circulation in the model is generally consistent with that inferred from the observed water properties (temperature, salinity, oxygen, and the stable oxygen isotope ratio, $\delta^{18}$O, Hirano et al., 2020). These patterns of the ocean properties and circulation are persistent throughout the year, but with a smaller magnitude of the warm water intrusion in autumn and winter (Fig. 11b).

## 4.2 Seasonal changes in Ekman downwelling and density surfaces

Ohshima et al. (1996) pointed out that the seasonal change in the alongshore wind controls the seasonality of the surface WW thickness in LHB (i.e., thickening in autumn and winter and thinning in spring and summer). A recent observational study by Hirano et al. (2020) indicated the weakening of the alongshore wind in summer allows the thickening of the CDW in the bay. Here, to confirm the wind-driven ocean processes, we examine the seasonality in ocean density surfaces and temperature in the deep depression of the bay. A control box was defined just south of the sill on the shelf break to estimate the seasonal cycle of ocean temperature and density profiles (Fig. 12). The box encompasses the observation station G3 (Fig. 8). As shown in the previous figures (Figs. 9 and 11), there is relatively warm water in the deep layer below the cold surface layer. The two-layer structure persists throughout the year in the mouth of the bay. The potential density surfaces of 27.5 kg m$^{-3}$ and 27.6 kg m$^{-3}$ fluctuate at approximately 400-m and 500-m depths, respectively, with a shoaling in summer and a deepening in autumn and winter. The seasonal differences of the density surfaces are up to 100m. As suggested in the literature (Hirano et al., 2020; Ohshima et al., 1996), this seasonal cycle of the density surfaces in the box area is explained by the seasonality of the alongshore wind stress (Fig. 2). Figure 13 shows a seasonal cycle of the monthly Ekman downwelling calculated from the alongshore wind stress. Here, the Ekman upwelling/downwelling velocity (negative values for downwelling) was calculated by $\tau$ ($\rho$ f $\Delta L$)$^{-1}$, where $\tau$ is alongshore (southwestward, see the unit vector in Fig. 3i) wind stress (but ignoring the existence of sea ice in this calculation); $\rho$ is ocean density; f is the Coriolis parameter at 68.5°S; $\Delta L$ is the assumed horizontal scale of the downwelling. We used $\Delta L = 25$ km for the calculation, and the width is approximately three-fold of the internal Rossby radius (8 km). We admit that the choice of $\Delta L$ is arbitrary, but the calculated Ekman downwelling can reasonably explain the seasonality of the density surfaces, particularly in the FI case. The seasonal cycle in the FI case is more apparent than that in the NOFI case because the fast ice cover plays a role as a physical barrier for the

surface momentum flux and the wind stress curl input caused by the combination of fast ice edge and the alongshore easterly wind result in pronounced Ekman downwelling in this region (Ohshima, 2000).

### 4.3 Basal melting at SGT

The warm water intrusion along the eastern flank of the deep trough (Figs. 9 and 11) results in a high basal melt rate at SGT

(Table 1 and Figs. 6 and 7). As an example, looking at the SGT melt rate distribution in the NOFI case (Fig. 8), active basal melting with the annual melt rate higher than 10 m yr$^{-1}$ is represented in the northern part of the SGT, where the warm water contacts first the SGT base. The southern part also has a high basal melt at approximately 5 m yr$^{-1}$. Since the circumpolar-averaged basal melt rate of the Antarctic ice shelves was estimated to be in a range from 0.81 to 0.94 m yr$^{-1}$ (Depoorter et al., 2013; Rignot et al., 2013), we can conclude that SGT is a hot spot of ocean-ice shelf/glacier interaction caused by the CDW

intrusions across the shelf break (Table 1).

Next, we examine the relationship of the seasonal cycles between the water mass transport into the SGT cavity and the monthly basal melt rate at SGT in the FI and NOFI cases (Fig. 14). We calculated the inflow transport of water masses exactly across the SGT ice front and the mean temperature in potential density bins with an interval of 0.02 kg m$^{-3}$. In both

cases, relatively warm water with temperatures higher than $-0.5$°C is present in the denser classes throughout the year, and the transport of the warm waters into the cavity reaches a maximum in summer (from November to February), resulting in the high basal melt rate at SGT. There are differences in the temperature and the total inflow volume of the warm water into the cavity in the two cases. The temperature of the inflow in the FI case is higher than that in the NOFI case, mainly because sea ice production in the southern part of LHB (i.e., the fast ice cover region in the FI case) in the NOFI case leads to cooling

of the surface-intermediate waters, particularly in autumn and winter (Figs. 4, 5, and 14). While the total transport of the inflow in the FI case is relatively stable throughout the year, there is a substantial seasonal cycle of the total inflow transport in the NOFI case. Integrated effects of the surface fluxes over the southern LHB region (e.g., sea ice production and wind stress on the ocean surface) that are missing in the FI case are likely to cause the pronounced seasonal cycle of the total inflow transport. In both cases, the temperature of the inflowing water masses starts to decrease in March (and the total

inflow volume transport declines in autumn and winter in the NOFI case), leading to a rapid decrease in ocean heat flux into the SGT cavity. Consistent with the seasonal cycle of the warm water intrusion, SGT shows a substantial seasonal variation of the basal melt rate from 12 m yr$^{-1}$ in winter to 18 m yr$^{-1}$ in summer in the FI case and 6 m yr$^{-1}$ in winter to 13 m yr$^{-1}$ in summer in the NOFI case (Fig. 7). It should be noted again that the melt rate in winter, even in the NOFI case, is still much higher than the circumpolar-averaged basal melt rate. The high basal melt rate throughout the year is caused by the persistent

warm water intrusion in the deep layer in the LHB region.

Hirano et al. (2020) used an ice radar of autonomous phase-sensitive radio echo sounder (ApRES) to estimate basal melt rate at SGT for the period from February-2018 to January-2019 (green line in Fig. 7). Here, we briefly compare the seasonality of the SGT basal melt rates between the model result and the observation-based estimate. The model and

observation both show the high basal melting rate in summer and the relatively low melt rate in autumn and winter (Fig. 7). The SGT-average basal melt rates in the model show a rapid increase from September to December, consistent with the ApRES estimate. These general agreements provide some confidence for the model results in this study, especially for the seasonality of the SGT basal melt rate. It should be noted that the ApRES located on the high-gradient zone of the basal melt rate in the model (Fig. 8) and the observation period was less than one year. When comparing the local basal melt rate

between the model and observation, the model underestimates the seasonal amplitude of the basal melt rate. This indicates the model underestimates the seasonal cycle of the warm water inflow to the cavity under the ApRES position. The bottom

topography under the SGT is outside of our compiled topography data, and thus it was not well constrained by the observed topography. The uncertainty of the bottom topography probably leads to the discrepancy in the seasonal amplitude of the local basal melt rate.


As shown in Section 3, a series of numerical experiments with different exchange coefficients under the fast ice shows different meltwater flux from the fast ice base (Table 1), and results from these experiments allow us to examine the impact of the fast ice melting on the SGT basal melting. The difference of the SGT melting among the experiments with the fast ice cover (FI and FIMOD series) is much smaller than that between the FI and NOFI cases. This is because the buoyant

meltwater from the fast ice mainly affects the surface water, and the SGT draft is deeper than the depth of the modified surface water. This explanation also holds for neighbor glaciers, the Skallen and Kaya Glaciers (Table 1).

**4.4 Warm water intrusions from the continental shelf break and slope regions to LHB**

The previous analyses in this section demonstrated that the warm water intrusions onto the continental shelf predominate the

ocean structure in LHB (Figs. 8, 9, and 11), the water masses flowing into the SGT cavity (Fig. 14), and the magnitude of the SGT basal melting (Figs. 6 and 7). Here, we examine in detail water mass exchanges across the sill section through the submarine canyons and ocean flow on the upper continental slope regions.

Firstly, we show vertical profiles of the potential temperature, potential density, and north/southward ocean flow along the

sill section at the latitude of 68.5°S to identify locations of the warm water intrusions from the offshore region to LHB (Fig. 15 for the NOFI case) and show the seasonal variation of the southward inflow transport across the sill section in the FI and NOFI cases (Fig. 16) . There are two submarine canyons in the section (we call the western and eastern canyons as "WC" and "EC", respectively). Looking at the deep layers denser than 27.5 kg m$^{-3}$ (Fig. 15), southward flows are present in the two submarine canyons and bring very warm water (up to 1°C) into LHB. The southward flows of the deep warm waters ($\sigma_0 >$

27.5 kg m$^{-3}$, green squares in Fig. 16 showing the density boundary) are identified in the two canyons throughout the year in both cases (Fig. 16). Looking at the upper layers, there are a strong northward flow of less dense/cold water on the western flank of the WC (Fig. 15) and surface-intensified southward flow over the eastern flank of the EC in winter (Fig. 15b). The surface-intensified southward flow brings a large volume of cold water into the bay in winter, in particular through the EC (Figs. 16b and 16d).


Next, we show the spatial distribution of ocean flows at the surface and 500 m depth to identify the origin of the warm water mass flowing to LHB through the submarine canyons (Fig. 17). South-westward flows along the continental slope predominate surface flows throughout the year. This alongshore surface flow is strong in winter and weak in summer (panels a, c, e, and g in Fig. 17), which corresponds to the seasonality of surface wind magnitude in this region (Fig. 3i). Ocean flow

patterns at 500-m depth are quite different from the surface flow patterns. In summer, a strong eastward flow is produced on the continental slope between the 500-m and 2000-m isobaths. In this study, we use the term "undercurrent" to point out the eastward flow below the westward surface flow. Although the eastward undercurrent is found in both the FI and NOFI cases, the undercurrent in the FI case is more vigorous than in the NOFI case. A part of the undercurrent is redirected southward at the submarine canyons. The southward flow brings warm water in the deep layer into LHB (Fig. 17b, see also Figs. 15a and

12). The southward transports of the warm water through the two submarine canyons are comparable to each other (Fig. 16). In the winter, the undercurrent weakens in the FI case and almost disappears in the NOFI case. In the FI case, the eastward flows are found only on the western side of the sill section. In the NOFI case and the eastern part of the sill section in the FI case, the direction of the subsurface flow on the continental slope becomes westward, which is the same direction of the

surface flow and the wind. Also, in the winter, a part of the westward subsurface flow is redirected southward at the
submarine canyons, bringing warm water into the bay (Figs. 17d and 17h, see also Fig. 15b). In the FI case, the weak
eastward undercurrent also contributes to the southward flow across the WC (Fig. 17d).

In order to examine the detailed characteristics of the undercurrent on the upper continental slope, vertical profiles of ocean
temperature, density, and east-westward flow along a section at 35.75°E (see the black line in Fig. 17) are plotted in Figure
18 for the NOFI case, which shows a clear seasonality in the undercurrent. In January, the surface-intensified westward flow
is present in the depth range from the surface to 400 m, with the maximum speed larger than 6 cm s$^{-1}$. The eastward-flowing
undercurrents locate in just offshore of the surface flow core in the depth range from 200 m to 1000 m (in the potential
density range from 27.5 kg m$^{-3}$ to 27.75 kg m$^{-3}$), and the maximum speed is approximately 6 cm s$^{-1}$. The undercurrent is
attached to the bottom of the upper continental slope in the water depth from 500 m to 700 m. Again, in July, the
undercurrent almost disappears in the NOFI case, and the surface-intensified westward flow becomes more vigorous,
extending the thickness to 1000-m depth. Figure 19a shows the seasonal variation of the eastward undercurrent transport in
the depth range 200–800m across the section (the sum of the only eastward transport). The transport reaches the maximum in
the spring and summer seasons from October to March and the minimum in the autumn and winter seasons from April to
August. In summary, the eastward-flowing undercurrent on the upper continental slope is more developed during the spring
and summer months (Fig. 19a) when the easterly coastal winds are weakest (Fig. 3i). It should be noted that the magnitude of
meltwater flux from the fast ice does not strongly affect the seasonality of the undercurrent on the upper continental slope
regions (Fig. 19a).

## 5 Interannual variability in the model

In the model, there is substantial interannual variability in the ocean conditions in LHB and the SGT basal melting (Figs. 6
and 7). In this section, we examine the interannual variability and explore the physical links among the oceanic variables and
the SGT basal melting in the model. However, it should be noted that our analysis period is too short (11 years from 2008 to
2018) to examine very-low-frequency variations (e.g., decadal variability). Furthermore, there are no observation to validate
the simulated interannual variability. A comparison with the result from the CKDRF case allows us to confirm that the
interannual variability in the FI and NOFI cases is not due to the model's drift or inherent variability and that the
interannual-varying surface forcing causes this variability. Although the magnitudes of the basal melt rate in the FI and
NOFI cases are different, the temporal evolutions of the interannual variability of the basal melting are similar to each other
(Fig. 6). The interannual variability of the SGT basal melting is the largest in summer, with the standard deviation of up to 3
m yr$^{-1}$ (Fig. 7).

As shown in Section 4, the warm water intrusions from the shelf break cause the active basal melting at SGT. Here, we show
that the interannual variability of the ocean temperature in the depression (at the box) explains the temporal variability in the
SGT's basal melt rate. Since the seasonal amplitudes in the ocean temperature and the SGT basal melting are larger than the
interannual variability (Figs. 7 and 12), we calculated anomaly time series of the variables from the monthly climatology to
examine the temporal variations in detail. The monthly climatology is the monthly mean of the variables averaged over the
period 2008–2018. The upper panels in Fig. 20 show the vertical profile of the ocean temperature anomaly and density in the
FI and NOFI cases. There are distinct differences in the depth range from 200 m to 600 m. We vertically averaged the
temperature anomalies within this depth range to extract the temporal change in the subsurface temperature (solid lines in
Fig. 20c). The time series of the subsurface temperature anomaly has a statistically significant positive correlation with those
of the SGT basal melt rate. The correlation coefficients between them are r=0.86 and r=0.63 in the FI and NOFI cases,

respectively. The values are above the 99% significant level (of r=0.224). The results strongly indicate that the interannual variability of the warm water in LHB is responsible for that in the SGT basal melting.

Looking at the time series of the vertically-averaged ocean temperature and the SGT basal melting anomalies (Fig. 20c), these anomalies fluctuate around zero in the period from 2008 to the first half of 2012. After that, the negative anomalies persist in the period from the second half of 2012 to the end of 2014. Subsequently, the ocean temperature and basal melting anomalies are swung to positive phases in the following period from 2015 to 2018. The vertical displacement of ocean density surfaces partially accounts for the time series of the ocean temperature anomaly: for example, the pronounced depression of the density surfaces in 2013–2014 (e.g., the potential density surface of 27.6 kg m$^{-3}$) is corresponding to the negative temperature anomaly. This indicates that cold surface water (i.e., WW) is depressed to deep layers, resulting in the temperature reduction in the intermediate depths. Note that in the NOFI case, interannual variability in sea ice production in LHB (Fig. 5b) also contributes to that in the ocean temperature anomaly.

However, the positive (warm) anomalies in the period 2015–2016 do not seem to be explained by changes in the density surfaces. We consider that changes in the ocean flow over the upper continental slope are likely responsible for the ocean temperature variability in LHB through the inflow across the sill section. Here, we utilized the strength of the eastward undercurrent on the upper continental slope as a proxy of the ocean variation outside of the bay because warm water (> 0.0$^{\circ}$ C) on the continental slope is present in the layers denser than 27.5 kg m$^{-3}$ (approximately 400–500 m in depth) and the core of the undercurrent exists in the same density/depth range (Fig. 18). In the same manner for the ocean temperature and SGT basal melting, we calculated the anomaly time series of the eastward transport of the undercurrent (Fig. 19b) to remove the seasonal signal and compared it with the ocean temperature anomaly. The interannual variability of undercurrent transport is as large as the seasonal cycle. In the period 2015–2016, the undercurrent transport anomaly is in a pronounced positive phase, indicating the larger eastward volume transport anomaly. As shown in Figs. 15–17, the ocean flows on the continental slope directly links to the warm water inflow toward LHB, and the undercurrent variability is also a candidate to explain the time series of the ocean temperature anomaly. In fact, the correlation coefficients between the ocean temperature and the undercurrent transport anomalies are significantly positive (r=0.44 in the FI case and r=0.45 in the NOFI case). Furthermore, significant positive correlations are also confirmed between the SGT basal melting and the undercurrent transport anomalies (r=0.41 in the FI case and r=0.45 in the NOFI case). In summary, the ocean temperature anomaly at intermediate depths in LHB is controlled by a combination of the local perturbation of density surfaces and the ocean flow variability on the upper continental shelf, giving rise to variability in the SGT basal melting.

## 6 Summary and Discussion

We have carried out numerical experiments with a high-resolution ocean-sea ice-ice shelf model to perform a detailed investigation of the ocean conditions in LHB and the basal melting at SGT. In our numerical modeling, we utilized the recently-refined local bathymetry in LHB, which includes multi-beam surveys from the JARE expeditions and depth information at control points from the Japan Coast Guard (Figs. 1 and 2). Our model can realistically capture the observed two-layer structure in LHB, consisting of cold-fresh water in the surface layer and warm-saline water in the deep layer (Fig. 9). The warm water in the deep layer is originated from the Circumpolar Deep Water on the continental slope regions. It intrudes from the submarine canyons at the shelf breaks (Figs. 8, 15, and 17), extending southward to the SGT cavity along the eastern flank of the north-southward running deep trough (Fig. 11). The warm water intrusions are persistent throughout the year, reaching the maximum in summer and the minimum in autumn and winter. The persistent warm water intrusions

into the LHB region cause active basal melting at SGT (Figs. 6, 7, and 14). The annual mean melt rates of SGT were estimated to be in a range from 8.8 m yr$^{-1}$ in the NOFI case to 15.2 m yr$^{-1}$ in the FI case (Table 1), which are much higher than the satellite-derived estimates of the circumpolar-averaged basal melt rates (Depoorter et al., 2013; Rignot et al., 2013). The predominant easterly wind in the coastal region (Fig. 3) controls the thicknesses of the cold surface and the warm deep waters (Fig. 12) through the seasonal change of Ekman downwelling (Fig. 13), and the changes in the magnitude of the warm water intrusion regulate both the seasonal and interannual variability of the SGT basal melting (Figs. 12, 14, and 20). The modeled seasonal cycle of the SGT-average basal melting is consistent with the seasonality of the SGT basal melting derived from the direct oceanographic and ice-radar observations (Fig. 7 and Hirano et al., 2020), but the local basal melt rate is underestimated.

The model results showed the large seasonality of the subsurface currents under strong westward surface flows over the continental slope regions, i.e., the clear appearance of the eastward undercurrent in summer (Figs. 17, 18, and 19). The subsurface flows play a role in the warm water mass inflow through the submarine canyons on the sills to LHB (Figs. 15, 16, and 17). We consider that this modeled flow pattern is a part of the Antarctic Slope Front (ASF)-Antarctic Slope Current (ASC)-Antarctic Slope Undercurrent system (Thompson et al., 2018) in this region. The ASF is a ubiquitous frontal structure formed by the prevailing easterly winds in the Antarctic coastal regions (Fig. 3), and the ASC is the corresponding strong westward-flowing ocean jet formed over the continental slope. The Antarctic Slope Undercurrent is established on the seafloor in some Antarctic continental slope regions, satisfying the thermal wind relationship. The magnitude of the eastward-flowing undercurrent off the LHB region is greatest in the spring and summer months (Fig. 19a) when the easterly coastal winds are at their weakest (Fig. 3i). In fact, the Antarctic Slope Undercurrent had been observed in the eastern Weddell Sea (Chavanne et al., 2010; Heywood et al., 1998; Núñez-Riboni and Fahrbach, 2009), in the Amundsen Sea (Walker et al., 2013), and in East Antarctica (Peña-Molino et al., 2016). These observational studies and the several modeling studies have pointed out that the seasonal and interannual changes of the Antarctic Slope Undercurrent play a pivotal role in water mass exchanges across the shelf breaks (Assmann et al., 2013; Silvano et al., 2019; Smedsrud et al., 2006). Until now, no observations have been made for the ocean current over the continental slope off the LHB region. If there is a seasonal undercurrent on the slope, LHB has a similar configuration to the Amundsen Sea, which has been well studied for ocean-ice shelf interaction. Since JARE performs observations within the LHB every year, LHB would be a useful monitoring site for ocean-ice shelf interaction in the East Antarctic region, although the total melting amount of the ice shelf/glacier is smaller than those of the ice shelves in the Amundsen Sea.

The existence of the multi-year fast ice characterizes LHB. The southern part of the LHB region is usually covered with a few-meters thick fast ice. We have examined the effects of the fast ice cover on the oceanic and sea ice fields and ice shelf/glacier melting, based on numerical experiments with and without the fast ice. In both experiments, the fluctuations in the alongshore wind are the main driving force for the seasonal-to-interannual variation of oceanic fields in LHB. It is found that fast ice plays a role as an insulator between the atmosphere and the ocean and suppresses local sea-ice production and cold water mass formation in the bay (Figs. 4, 5, 9, and 10). As a result, in the FI case, the intermediate-depth warm water intrusions from the shelf break regions reach the SGT cavity with little or no modification, resulting in higher basal melting at SGT (Figs. 7 and 14). These modeling results strongly suggest that along with the alongshore wind, fast ice in the LHB region is an essential factor controlling the ocean-ice shelf/glacier interaction, findings reported in the other regions of East Antarctica (e.g., Fraser et al., 2019; Massom et al., 2001). In this study, we assumed the shape of the fast ice unchanged in the experiments, however, in reality, there is interannual variation in the fast ice shape and extent (Fraser et al., 2012, 2020). Therefore, it should be kept in mind that the interannual variability in the fast ice extent is also a responsible factor for those in the ocean conditions in LHB and the SGT basal melting.

580

Although our study area, LHB, covers only a small portion of the Antarctic coastal regions, we have been able to perform integrated research, combining topographic surveys, ship-based oceanographic observation, satellite observation, and a regionally-high resolution ocean-sea ice-ice shelf modeling. There are still large uncertainties in the bottom topography in the Antarctic coastal regions (especially thick ice-covered areas: e.g., under the fast ice and ice shelf/glacier), and the

585 uncertainty directly links to the reliability of the numerical model results. Needless to say, further exploration of the Antarctic coastal bathymetry is required because the detailed bathymetry such as submarine canyons play a key role in the Antarctic coastal water masses and the subsequent ice-ocean interaction. We demonstrate that the high-resolution ocean modeling with the updated bottom topography can reasonably reproduce the observed oceanic field and the SGT basal melting, and such numerical modeling is useful to fill the spatiotemporal gaps in the observations. Combined with the in-situ

590 and satellite observations, numerical modeling is a very powerful tool for a better understanding of ocean-ice shelf interaction over the Southern Ocean.

**Acknowledgments:** KK was supported by JSPS KEKENHI Grants JP19K12301, JP17H06323. DH was supported by JSPS KEKENHI Grants JP20K12132, JP17K12811. MF was supported by JSPS KEKENHI Grants JP17H06322 and the Science Program of Japanese Antarctic Research Expedition (JARE). AF was supported by grant funding from the Australian Government as part of the Antarctic Science Collaboration Initiative program (grant ASCI000002). TT was supported by Grant-in-Aid for Scientific Research (17H04710, 17H06317 and 17H06322) from the MEXT of the Japanese Government, the Science Program of Japanese Antarctic Research Expedition (JARE) as Prioritized Research Project (AJ0902: ROBOTICA), National Institute of Polar Research (NIPR) through Project Research KP-303, the Center for the Promotion of Integrated Sciences of SOKENDAI, and the Joint Research Program of the Institute of Low Temperature Science, Hokkaido University. We express our deep gratitude to Dr. Yoshifumi Nogi for providing the bathymetry data for scientific study in Lützow-Holm Bay. The Japan Coast Guard officially provide the bathymetric data obtained onboard the Icebreaker Shirase during the JARE, including Lützow-Holm Bay. We are grateful to Dr. Shigeru Aoki and Dr. Keith Nicholls for providing the processed ApRES data. We appreciate two anonymous reviewers for their careful reading and constructive comments on the manuscript, and their comments were very helpful to improve the manuscript.

**Author contributions:** K.K led this study by conducting all the numerical experiments and analyses. All authors discussed the results and comments on the manuscript.

**Data availability:**

LHB bottom topography: http://dx.doi.org/10.17632/z6w4xd6s3s.1#file-845c7d35-fb34-435f-bfc8-57313b60ccbf

ETOPO1: https://www.ngdc.noaa.gov/mgg/global/global.html

ETOPO5: https://www.ngdc.noaa.gov/mgg/global/etopo5.HTML

GEBCO: https://www.gebco.net/data_and_products/historical_data_sets/#gebco_one

RTopo1: https://doi.pangaea.de/10.1594/PANGAEA.741917.

Rtopo2: https://doi.pangaea.de/10.1594/PANGAEA.856844

PHC: http://psc.apl.washington.edu/nonwp_projects/PHC/Climatology.html.

ERA-Interim: http://www.ecmwf.int/en/research/climate-reanalysis/era-interim.

All numerical experiments were conducted using COCO (https://ccsr.aori.u-tokyo.ac.jp/~hasumi/COCO/) with an ice-shelf component (Kusahara and Hasumi, 2013) and all the numerical model results for the plots in this study will be available in a data repository (http://dx.doi.org/10.17632/z6w4xd6s3s.1).

**Competing interest:** The authors declare no conflict of interest.

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

**Table 1:** Annual mean basal melt rate (m yr$^{-1}$) and amount (Gt yr$^{-1}$) at glaciers in the LHB and fast ice. SGT indicates the Shirase Glacier Tongue, SkG the Skallen Glacier, KG the Kaya Glacier.

| | Area | SGT | fast ice | SkG | KG |
|---|---|---|---|---|---|
| | extent (km$^2$) | 559 | 19.3×10$^3$ | 29.7 | 703 |
| Experiments | FI | 15.2 m/yr (7.8 Gt/yr) | 2.29 m/yr (40.7 Gt/yr) | 19.0 m/yr (0.52 Gt/yr) | 11.8 m/yr (7.6 Gt/yr) |
| | FIMOD05 | 14.5 m/yr (7.5 Gt/yr) | 1.50 m/yr (26.5 Gt/yr) | 18.0 m/yr (0.49 Gt/yr) | 11.1 m/yr (7.1 Gt/yr) |
| | FIMOD02 | 14.8 m/yr (7.6 Gt/yr) | 0.83 m/yr (14.7 Gt/yr) | 18.2 m/yr (0.50 Gt/yr) | 11.0 m/yr (7.1 Gt/yr) |
| | FIMOD00 | 14.3 m/yr (7.3 Gt/yr) | 0.0 m/yr (0.0 Gt/yr) | 16.6 m/yr (0.45 Gt/yr) | 10.0 m/yr (6.4 Gt/yr) |
| | NOFI | 8.8 m/yr (4.5 Gt/yr) | n/a | 7.5 m/yr (0.20 Gt/yr) | 4.5 m/yr (2.9 Gt/yr) |

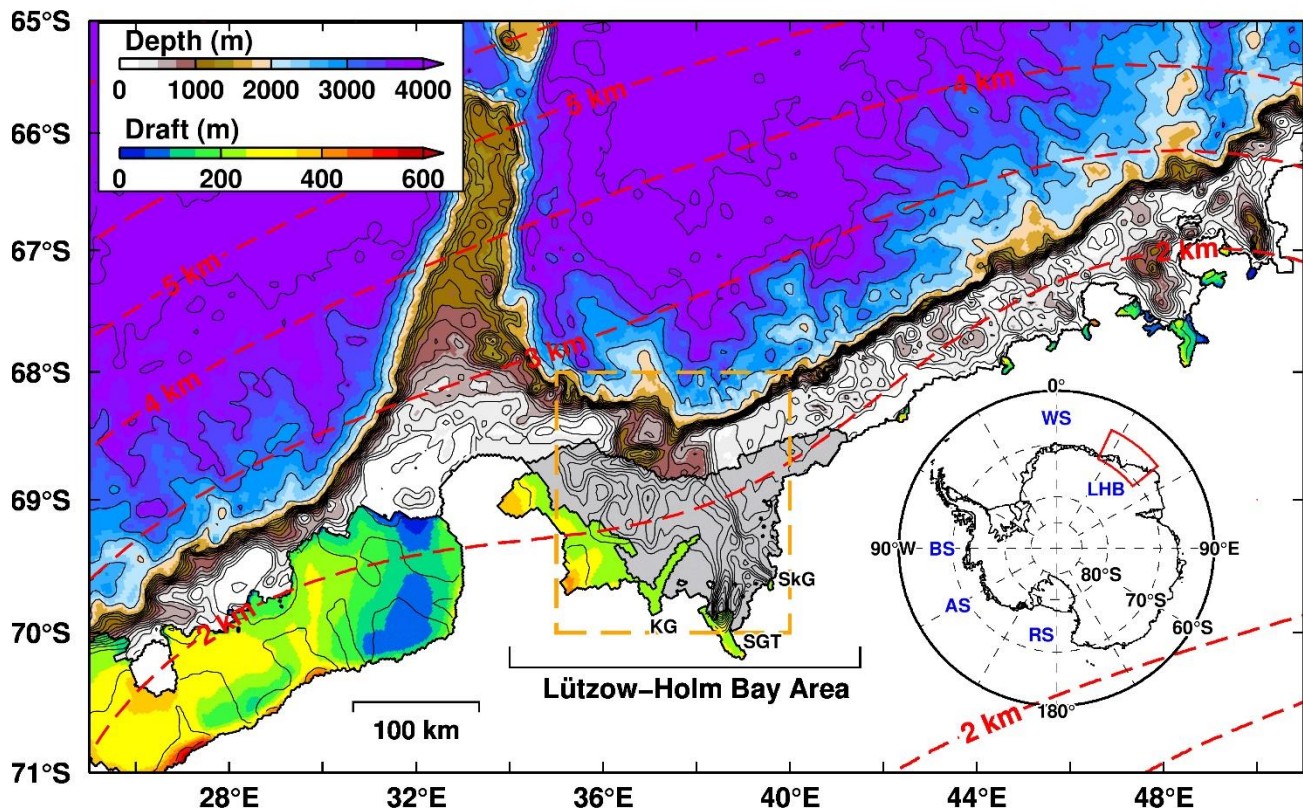

**Figure 1:** Bottom topography and the draft of ice shelves/glaciers around the Lützow-Holm Bay (LHB) area. Red dashed contours indicate the horizontal grid spacing in the model. Map inset shows our focal region in the Southern Ocean. Gray shaded areas are fast ice regions. Background bottom topography is derived from ETOPO1 dataset, and the local topography in the area enclosed with an orange box is replaced with the recently-compiled new dataset (section 2.2 and Fig. 2a). SGT indicates the Shirase Glacier Tongue, SkG the Skallen Glacier, KG the Kaya Glacier. Major place names are shown in the inset: Weddell Sea (WS), Ross Sea (RS), Amundsen Sea (AS), and Bellingshausen Sea (BS).

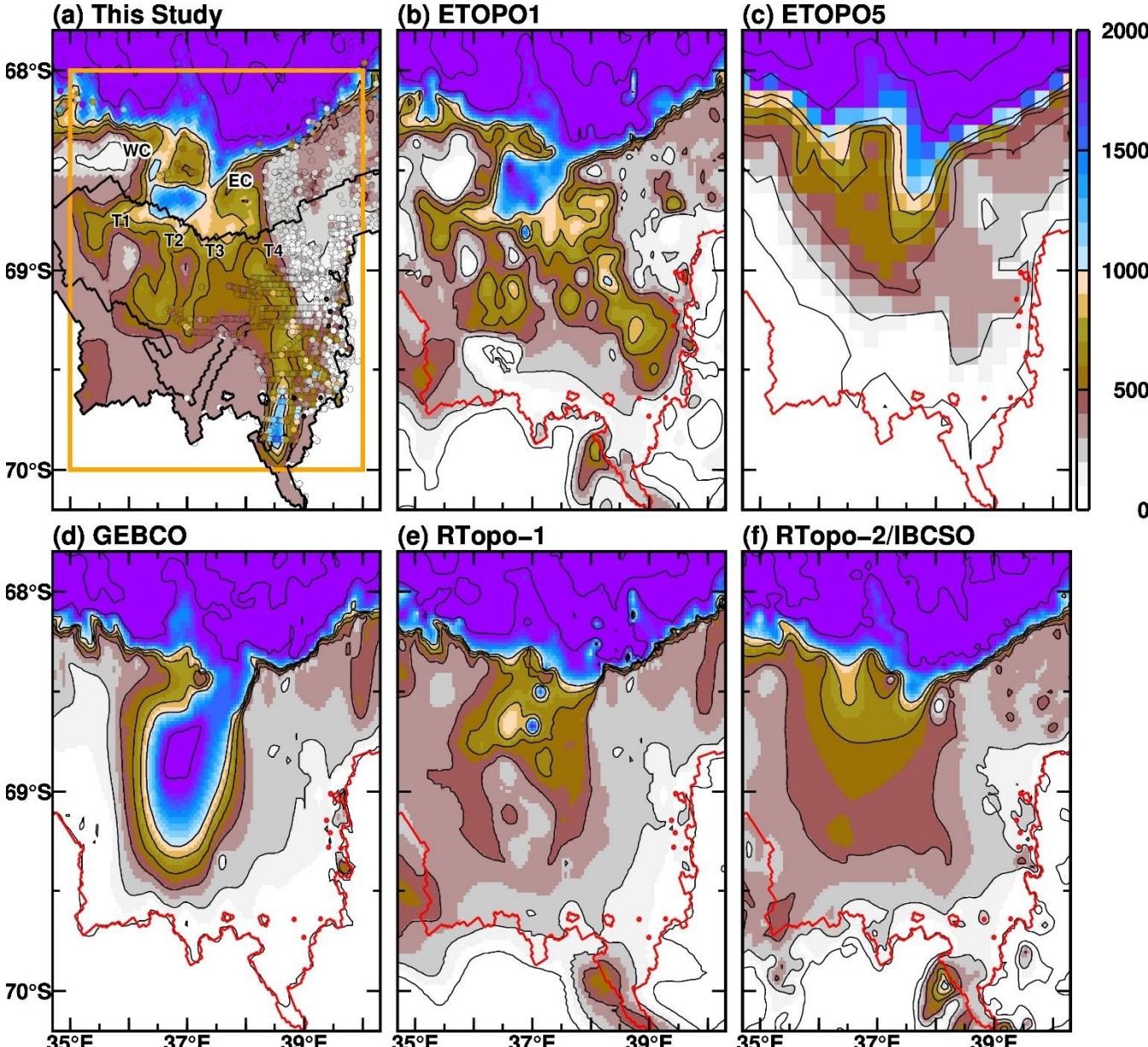

**Figure 2:** Comparison of water depth representation in the topographic datasets (a: this study, b: ETOPO1, c: ETOPO5, d: GEBCO, e: Rtopo-1, and f: Rtopo-2/IBCSO). The red lines in panels b–f indicate the grounding line in the model (a). Circles with color in panel (a) are observed depth measurements from the point echo sounding using sea ice drill holes and the JCB nautical chart (subsampled to avoid clutter in the panel). In panel (a), the two submarine canyons around the sill are labeled with WC and EC, and four troughs extending southward from the sill are marked with T1, T2, T3, T4 from west to east.

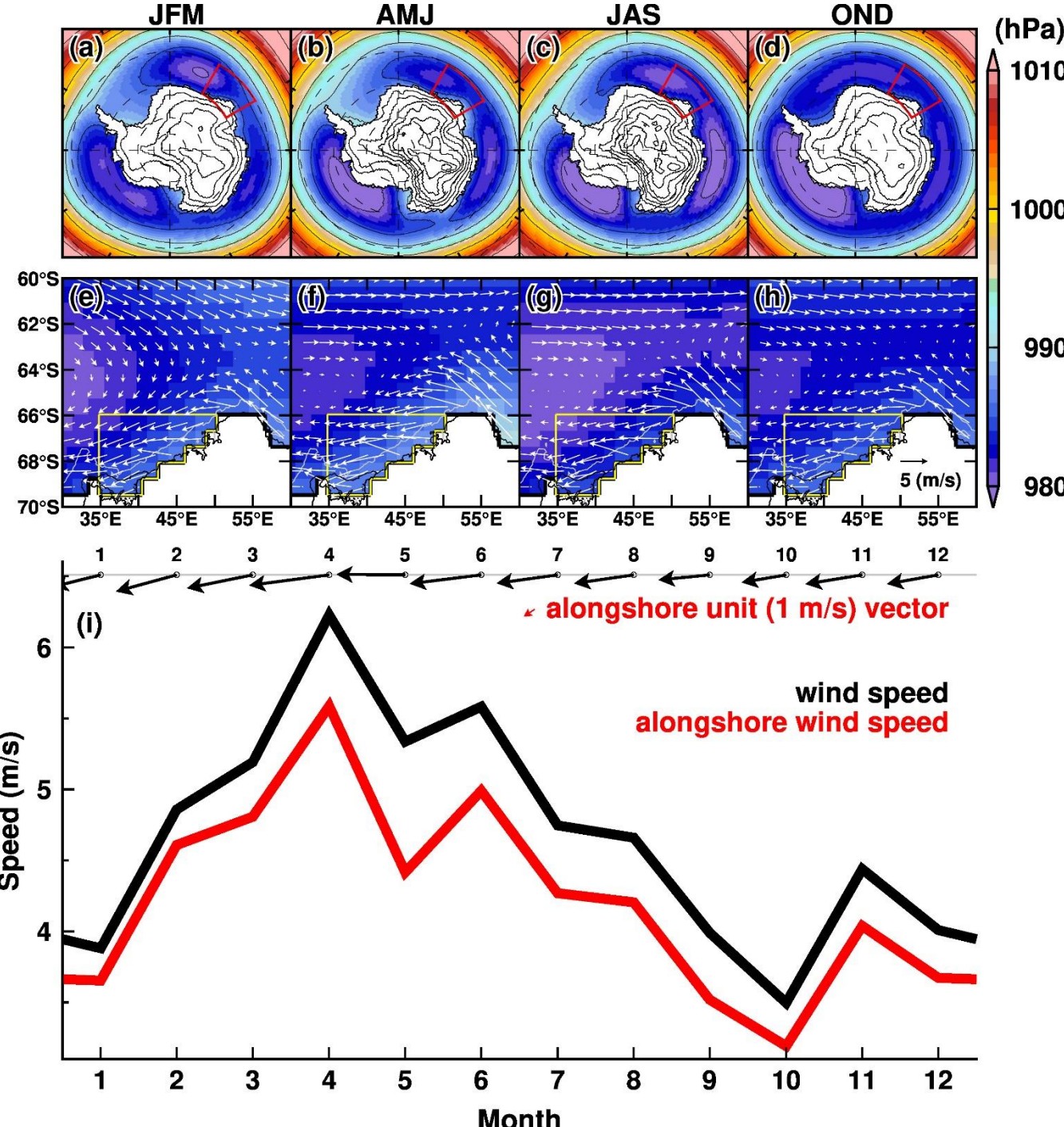

**Figure 3:** Seasonal climatology of atmospheric circulation (a–d) over the Southern Ocean and (e–f) in the region off Lützow-Holm Bay. Color and vectors show the 3-month average of surface air pressure and 10-m wind fields, respectively. The climatology is calculated from the ERA-Interim dataset for the period 2008–2018. The red box in the upper panels indicates the region for panels e–h. The area enclosed by the yellow line in the middle panels is the area for averaging the 10-m wind for panel (i). In panel (i), the monthly climatologies of wind vector and wind speeds (black: absolute speed and red: alongshore speed) are displayed. The red vector is a unit vector of the defined alongshore direction.

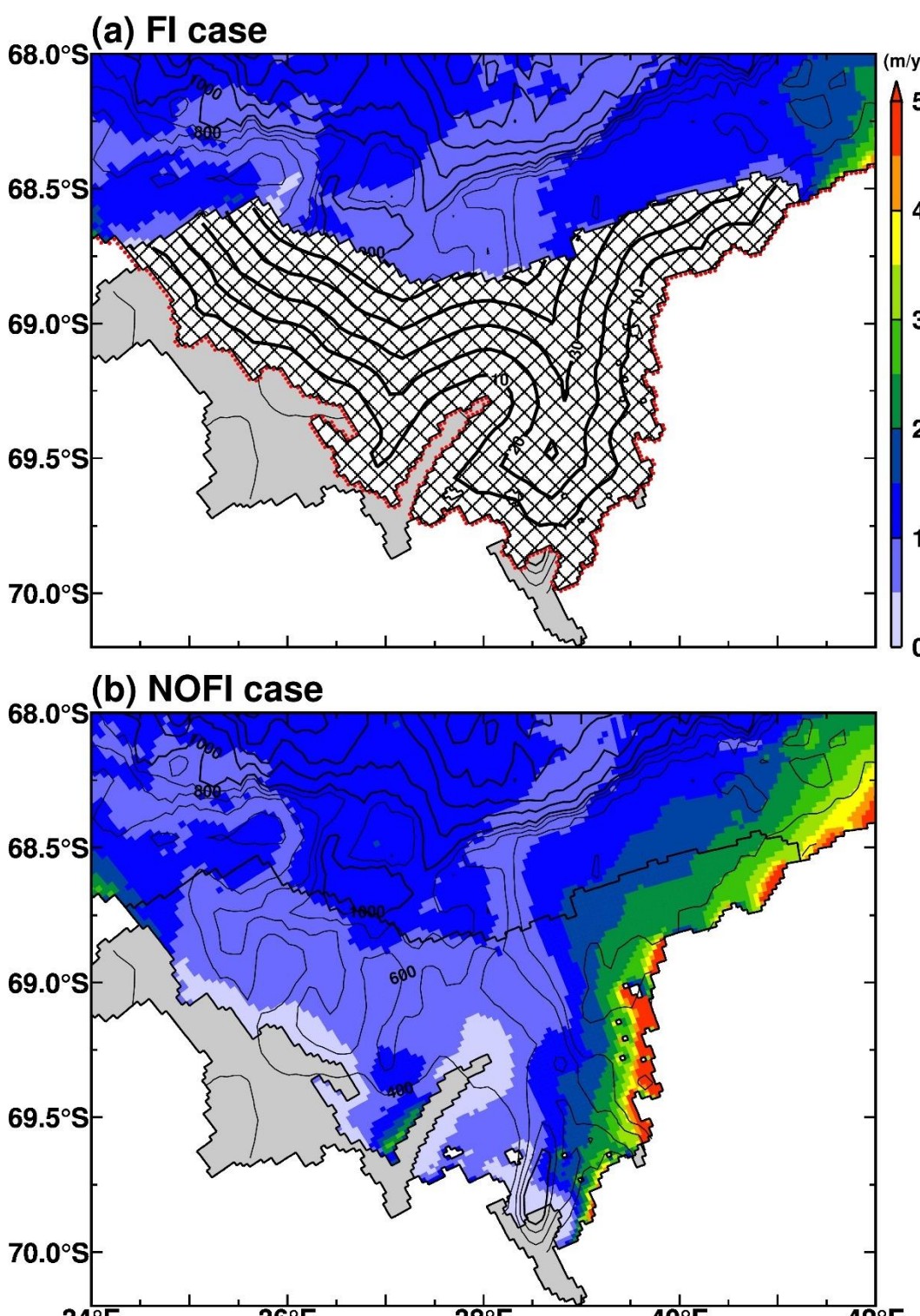

830 **Figure 4:** Maps of annual sea ice production (m yr$^{-1}$) in the (a) FI and (b) NOFI cases. The sea ice production was averaged over the period 2008–2018. The meshed region in panel (a) indicates the fast ice cover. Gray shaded areas show ice shelves/glaciers. The contours over the fast ice cover in panel (a) represent the minimum distance from the coastline or ice front (shown by red dots). The black contours show the water depth, with 200-m (1000-m) intervals in the regions shallower (deeper) than 1000 m.

835

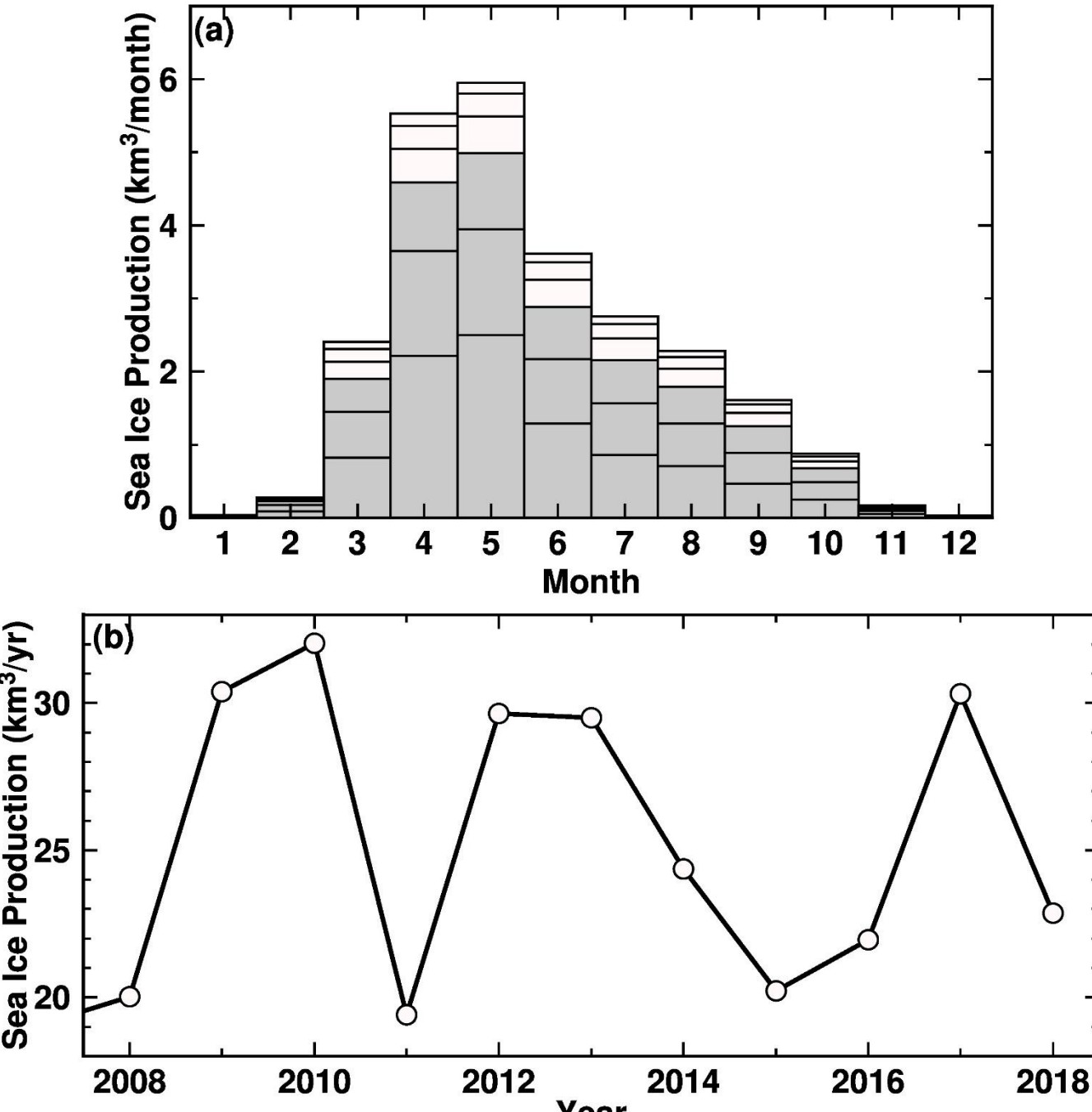

**Figure 5:** Seasonal and interannual variations of sea ice production in the LHB region in the NOFI case. Note that sea ice production is calculated from sea ice freezing, not including sea ice melting. In panel (a), the seasonality of the sea ice production was calculated in 10-km bin of the distance from coastline/ice front (see contours over the fast ice region in Fig 4a), and the grey boxes show the sea ice production within 30-km from coastline/ice front. The monthly climatology is the average over the period 2008–2018.

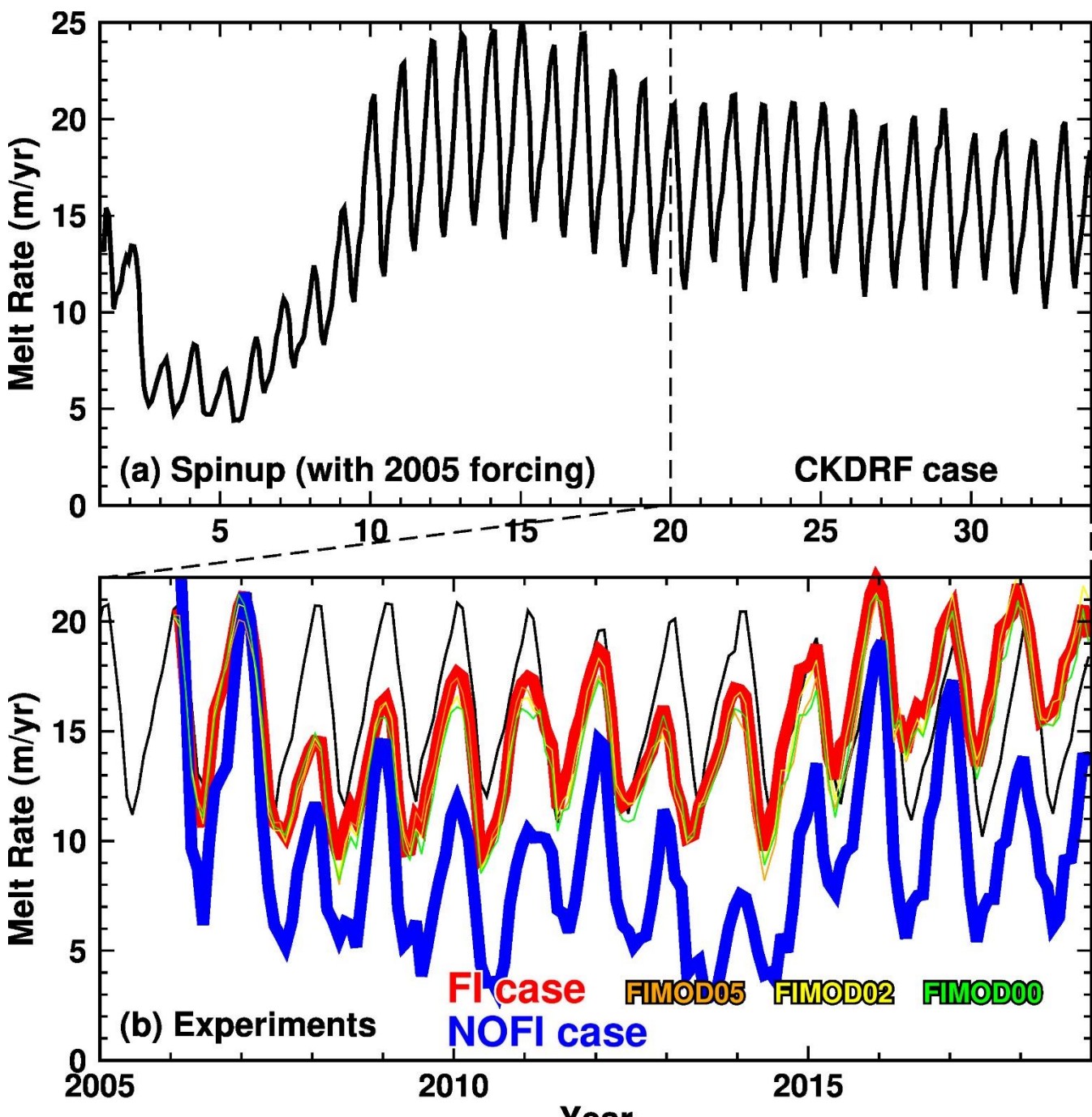

**Figure 6:** Time series of basal melt rate at Shirase Glacier Tongue (a: the full integration period of 33 years including periods of the initial 20-year spin-up and the CKDRF case, b: the last 14 years corresponding to the period 2005−2018). In panel (b), red, blue, and black lines show the results from the FI, NOFI, and CKDRF cases, respectively. Thin orange, yellow, and green lines are the results from the FIMOD05, FIMOD02, and FIMOD00 cases, respectively. The equivalent annual melt rate (m yr$^{-1}$) is used for the vertical scale.

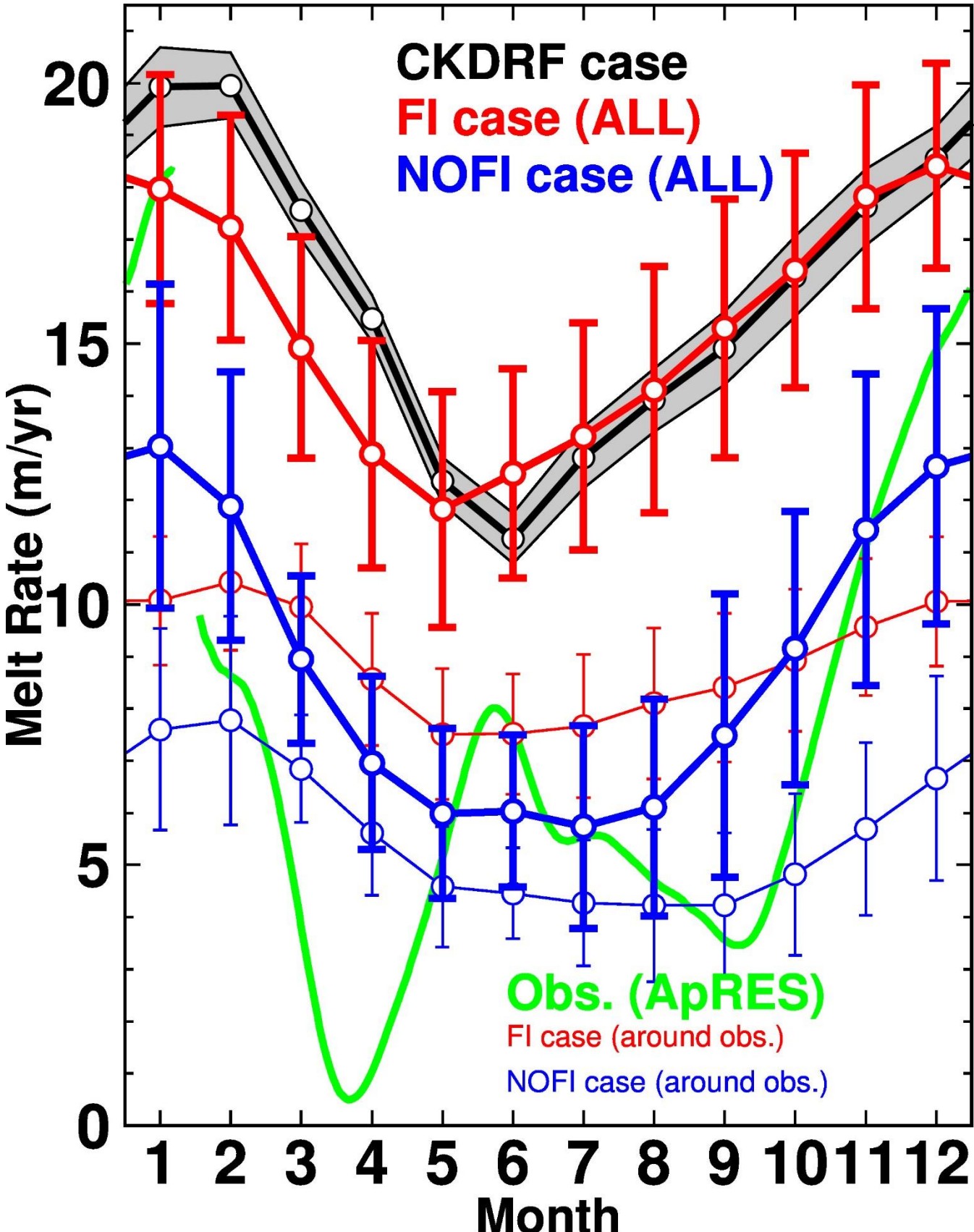

**Figure 7:** Seasonal variation of basal melt rate at Shirase Glacier Tongue. Black, blue, red, and green lines show results from the CKDRF, NOFI, FI cases, and ice radar-derived estimate (ApRES, Hirano et al. 2020), respectively. Thick lines represent the mean melt rate averaged all over the SGT. Thin red and blue lines show the regional basal melt rate around the ApRES observation (see Fig.8 for the locations). Shade or vertical bars indicate the standard deviation of the monthly basal melt rate in the period 2008–2018, showing the model's inherent variability and interannual variability. The scale of the vertical axis is the equivalent annual melt rate (m yr[-1]).

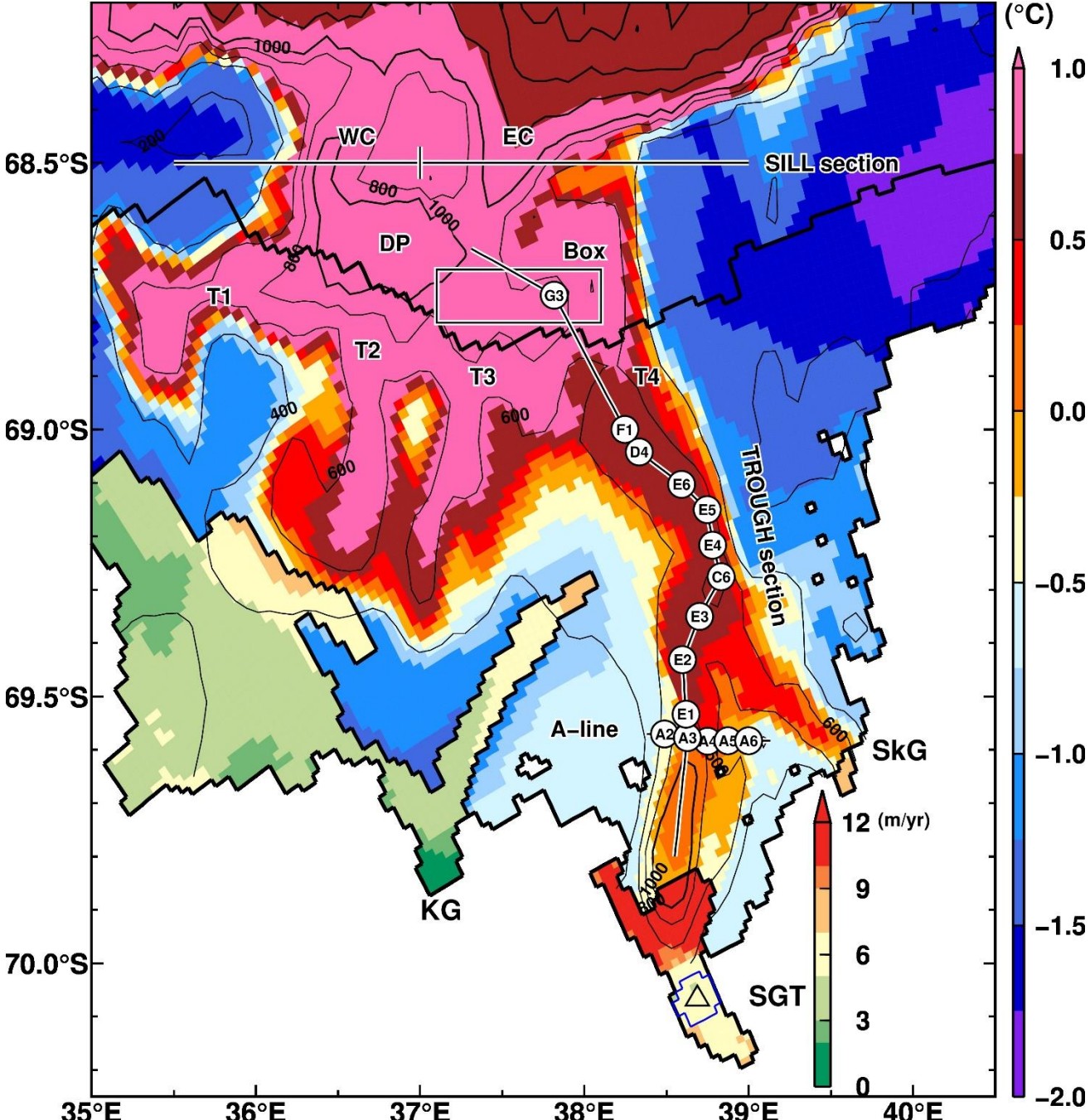

**Figure 8:** Key sections and observational stations used in this study. Colors represent the modeled ocean bottom temperature in January and the annual mean basal melt rate in the NOFI case. The variables were averaged over the period 2008−2018. White circles with station names show positions of the in-situ oceanographic observations for the comparisons in Figs. 9 and 11. Black lines connecting the observational stations are defined as TROUGH section and A-line. The thick zigzag black lines are the grounding line, the ice-front line, and the fast ice edge in the FI case. The area enclosed with the blue line on the SGT is the averaging area for calculating the regional basal melt rate around the location of ApRES (triangle). SGT indicates the Shirase Glacier Tongue, SkG the Skallen Glacier, KG the Kaya Glacier. WC and EC are placed to refer to the western and eastern submarine canyons, respectively. T1, T2, T3, T4 are placed to refer to the troughs in the LHB. Sill sections near the shelf break are used to show vertical profiles in Fig 9. A box is used for averaging ocean properties in Figs. 12 and 20.

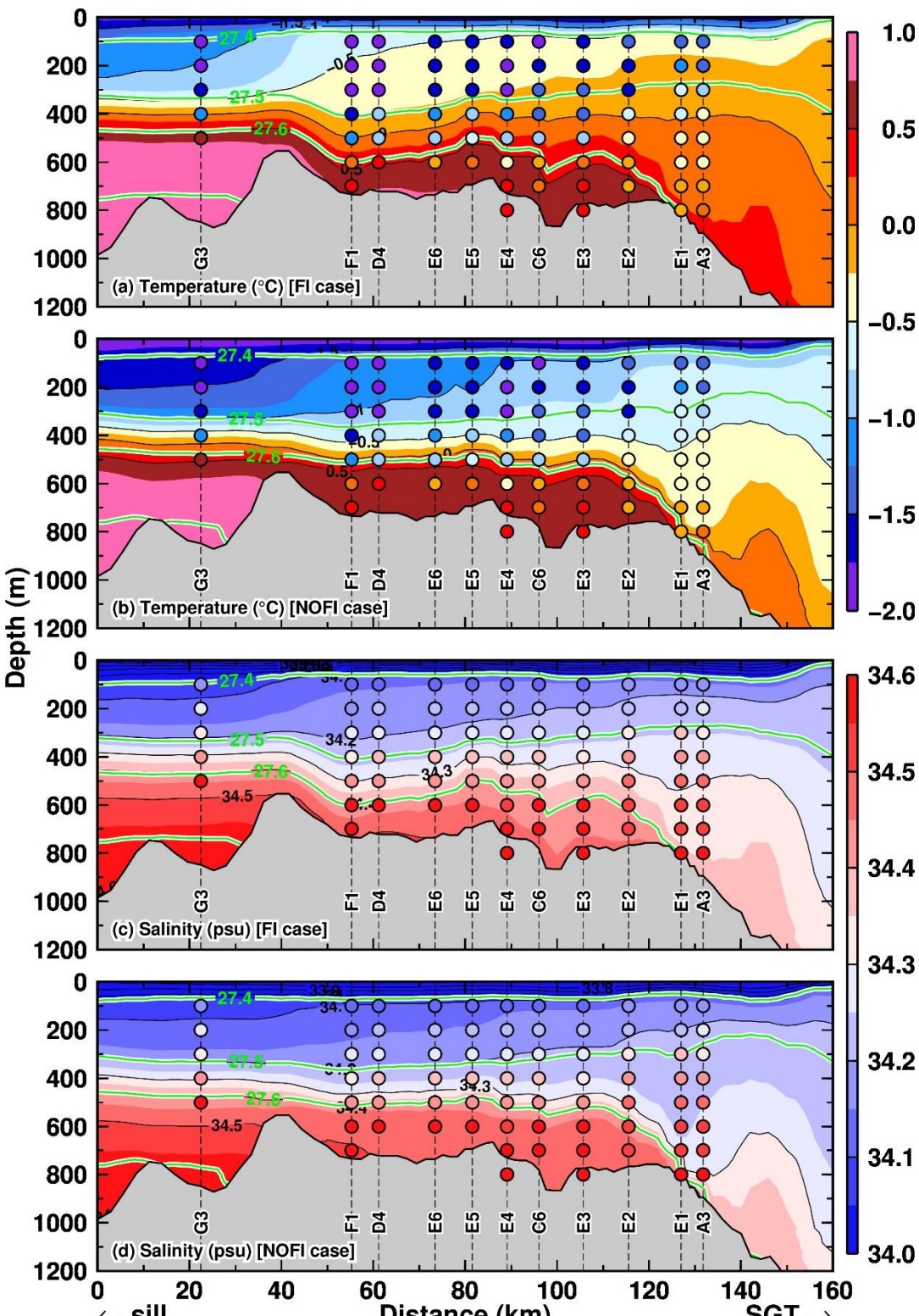

**Figure 9:** Vertical profiles of (a–b) potential temperature and (c–d) salinity along the TROUGH section in January in the two cases (a and c for the FI case, b and d for the NOFI case). Green curves indicate the contours of potential density anomalies. The model's variables were averaged over the period 2008–2018. The horizontal axis indicates the distance from the starting point of the TROUGH section located near the deepest depression in LHB (see the line of the TROUGH section in Fig. 8). Circles filled with the color at vertical dashed lines indicate the observed ocean properties in January/February 2016 (Hirano et al., 2020).

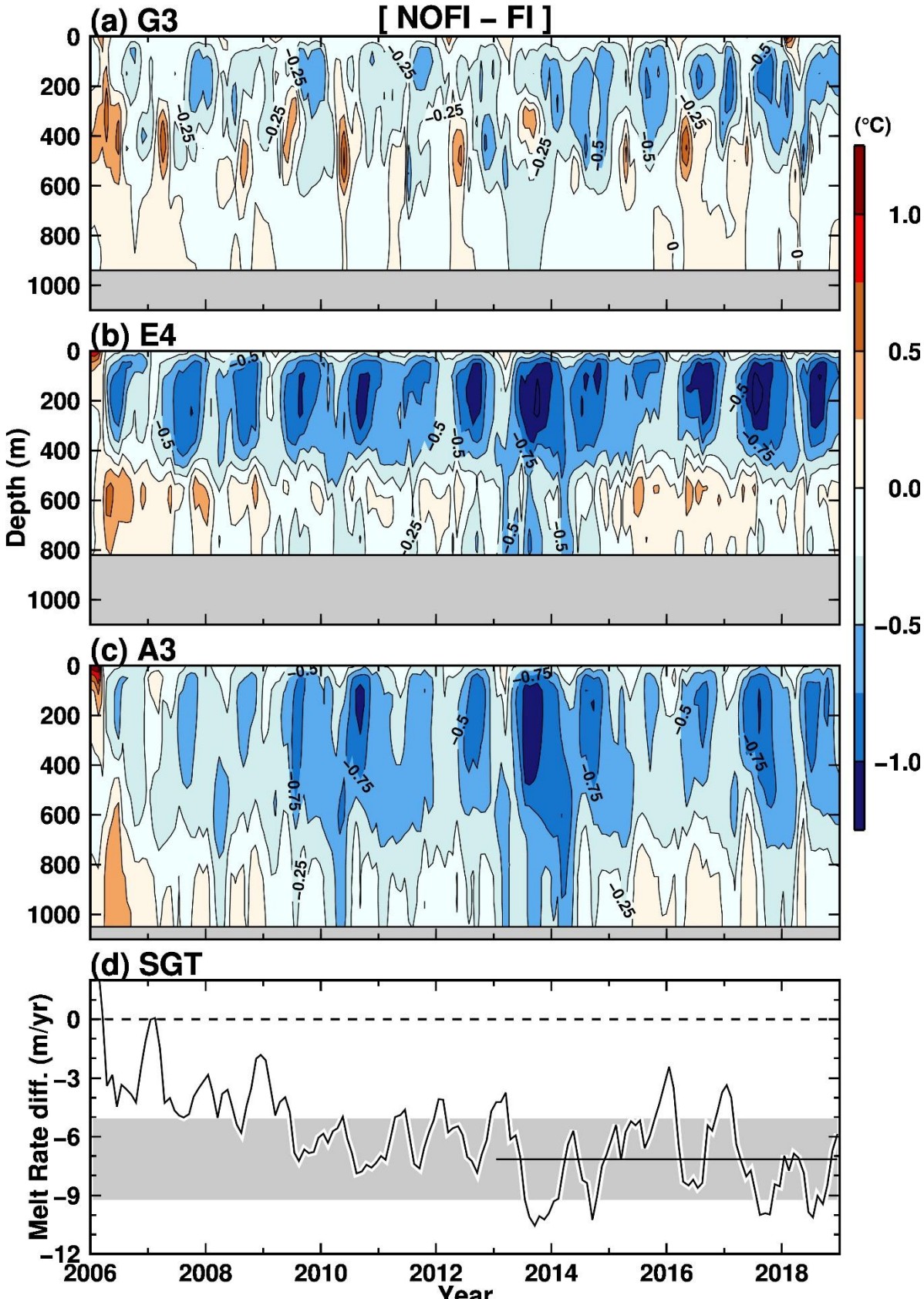

**Figure 10:** Time evolutions of the differences (NOFI − FI) in ocean temperature at the stations G3, E4, and A3 on the TROUGH section (Figs. 8 and 9), and the SGT basal melting. The black line and shade in panel (d) indicate the average of the SGT melt rate difference averaged over the period 2013–2018 between the FI and NOFI cases and the standard deviation.

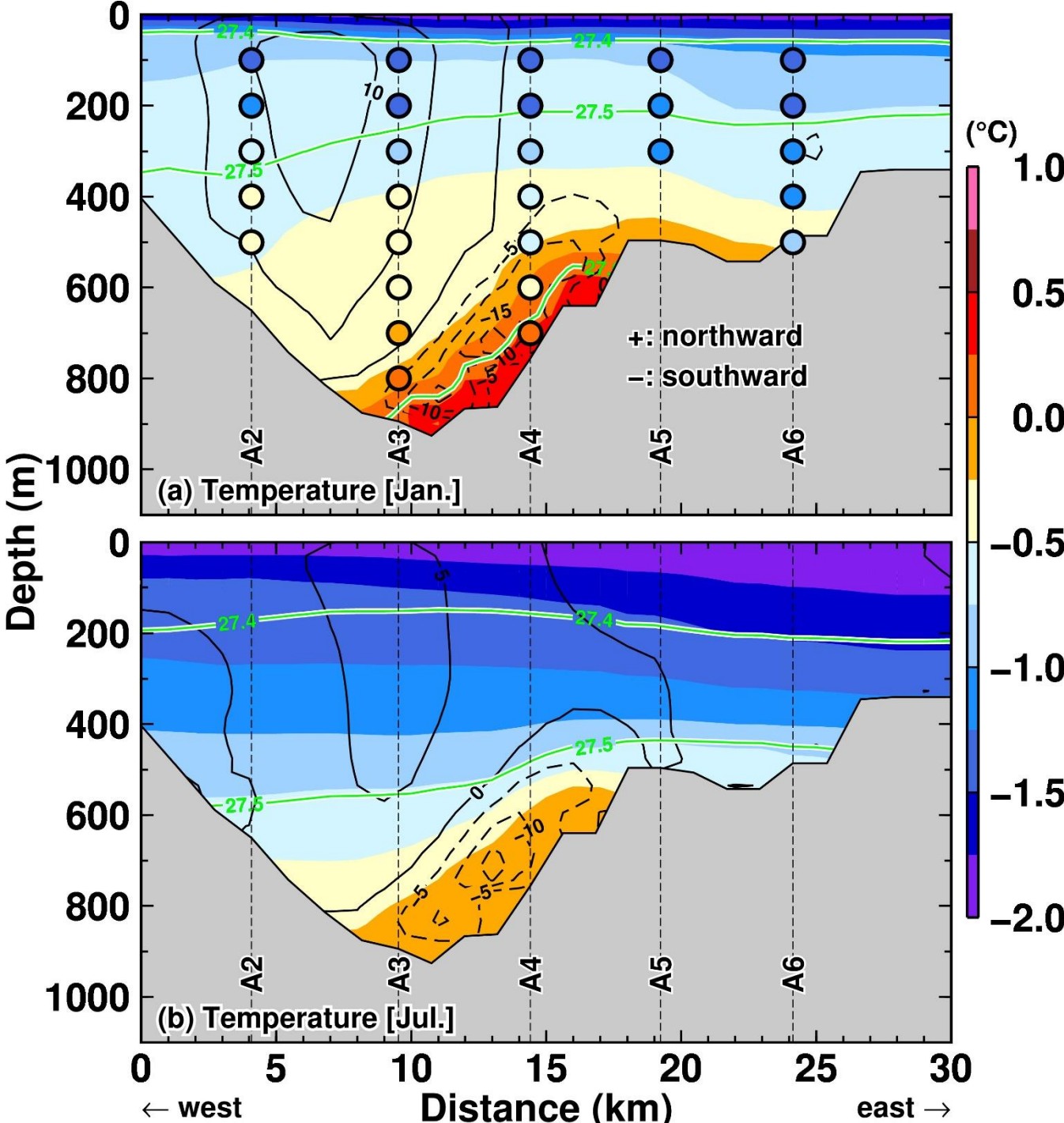

**Figure 11:** Vertical profiles of potential temperature along the A-line in (a) January and (b) July in the NOFI case, with the velocity normal to the section (cm s$^{-1}$). Positive velocity indicates northward flow. Green curves indicate the contours of potential density anomalies. The model's variables were averaged over the period 2008–2018. The horizontal axis indicates the distance from the starting point of the A-line section (see the line of the A-line in Fig. 8). Circles filled with the color in panel (a) indicate the observed ocean properties in January/February 2016 (Hirano et al., 2020).

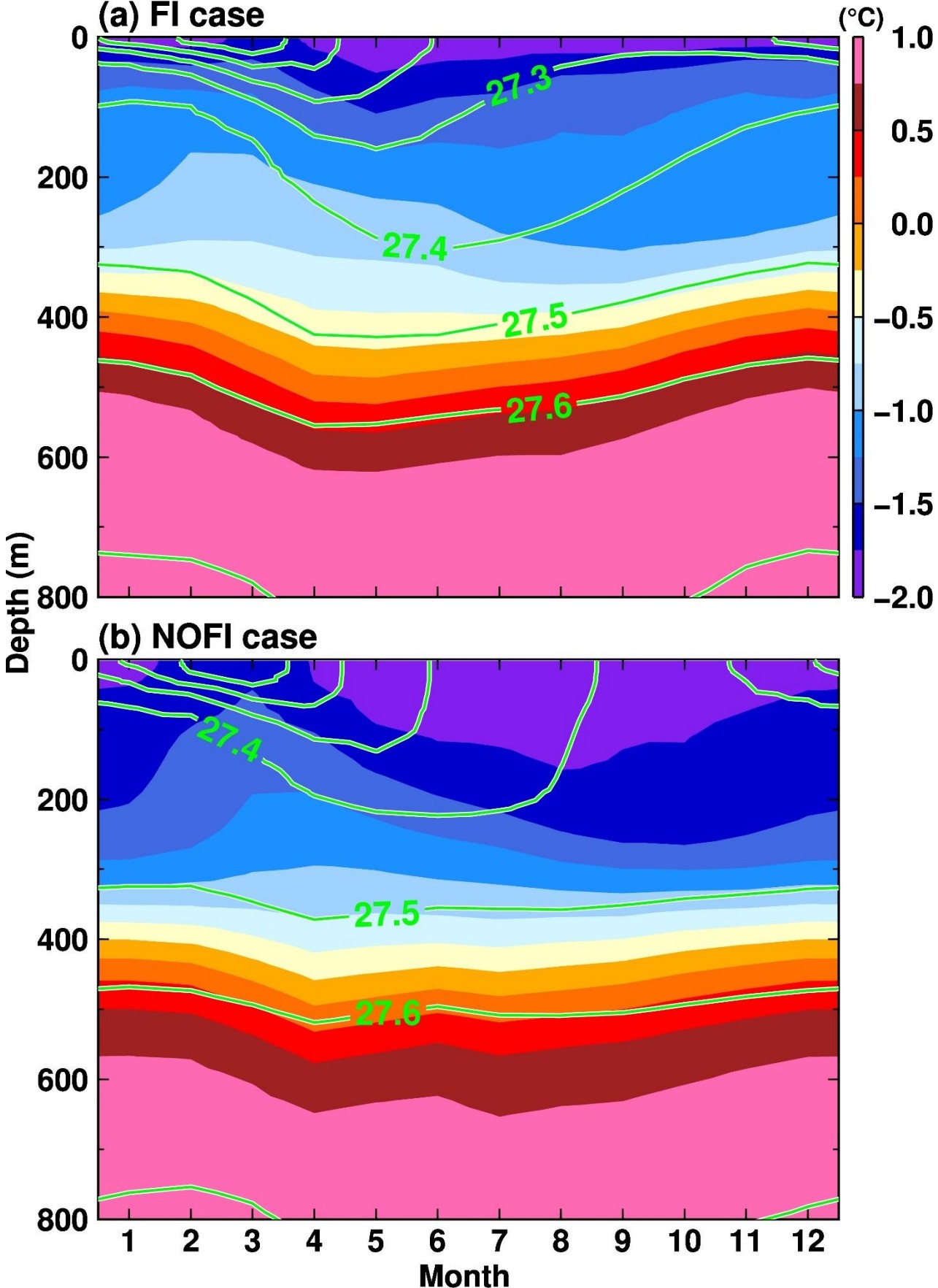

885 **Figure 12:** Seasonal variation of potential temperature (color) and potential density anomaly (green contours) averaged over the box area
(Fig. 8) in the (a) FI and (b) NOFI cases. The variables were averaged over the period 2008–2018.

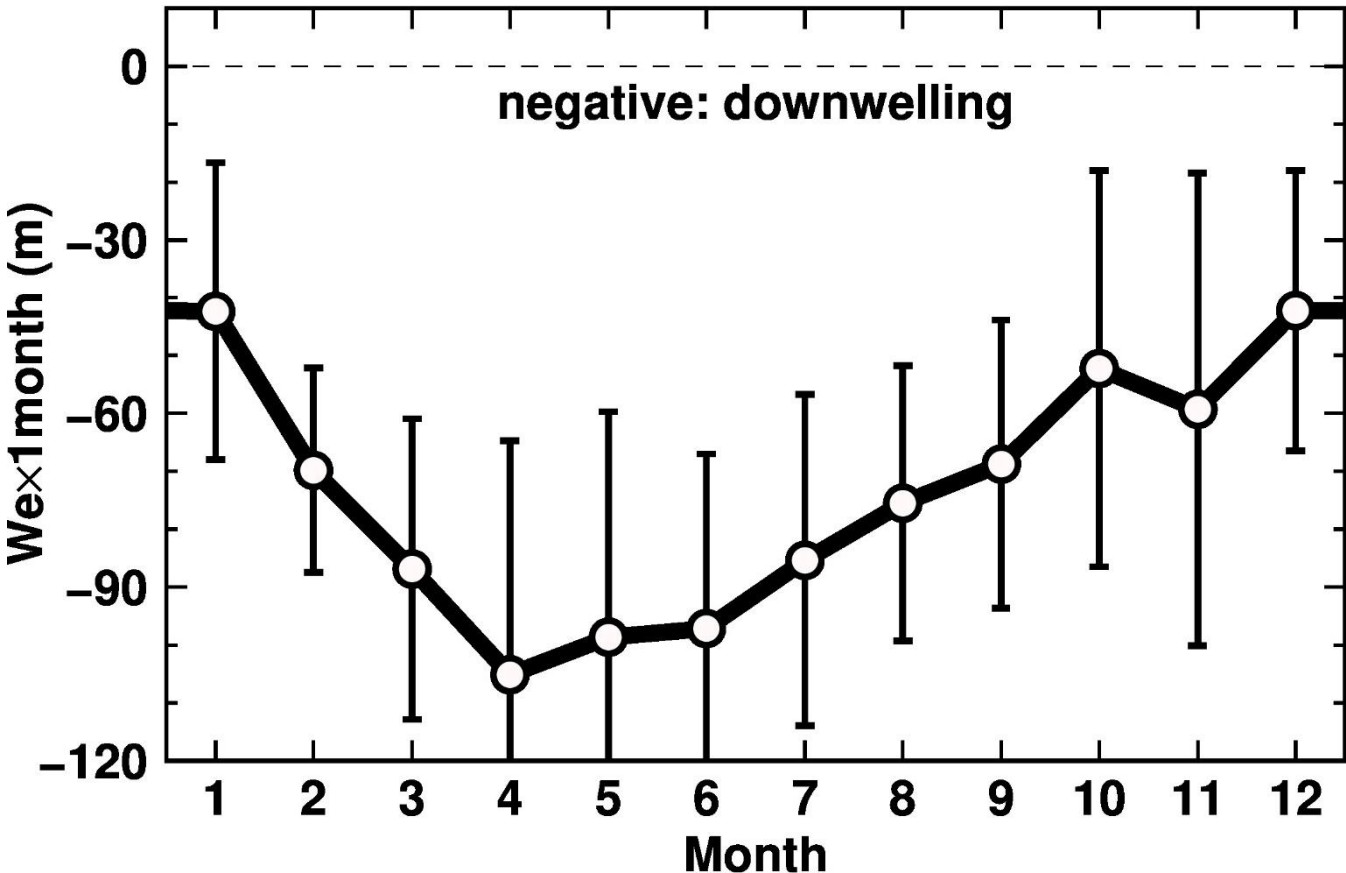

**Figure 13:** Seasonal variations of monthly Ekman downwelling (a product of Ekman velocity and a month (30 days)). The monthly value is climatology averaged over the period 2008–2018.

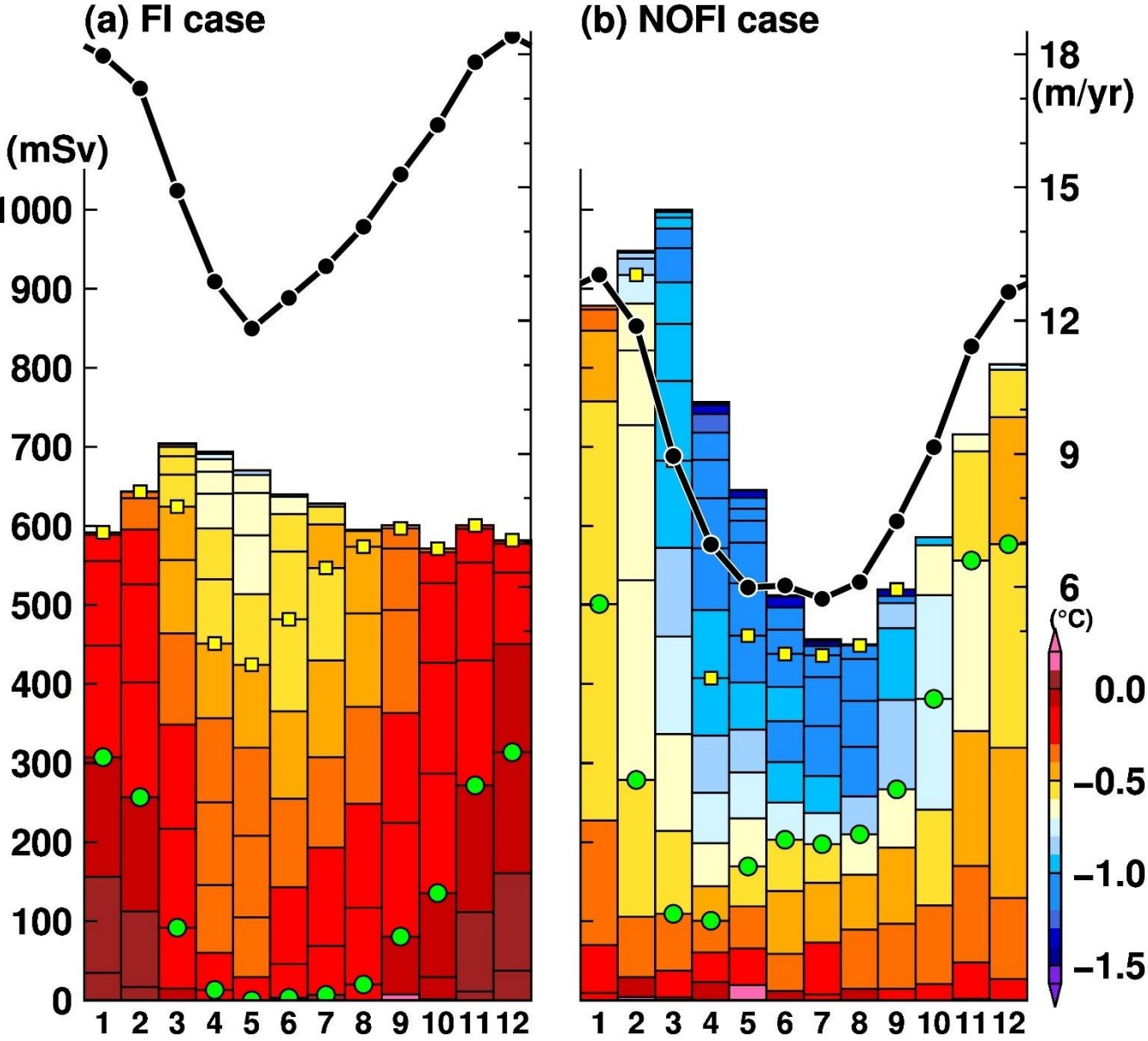

**Figure 14:** Seasonal variation of water mass transport flowing into the SGT cavity (mSv, left axis for boxes) with the potential temperature in potential density anomaly bin with 0.02 kg m$^{-3}$ (color) and the mean basal melt rate (m yr$^{-1}$, right axis for black lines) in the (a) FI and (b) NOFI cases. Green dots and yellow squares represent boundaries of 27.5 and 27.4 kg m$^{-3}$ of the potential density anomaly, respectively. The variables were averaged over the period 2008–2018.

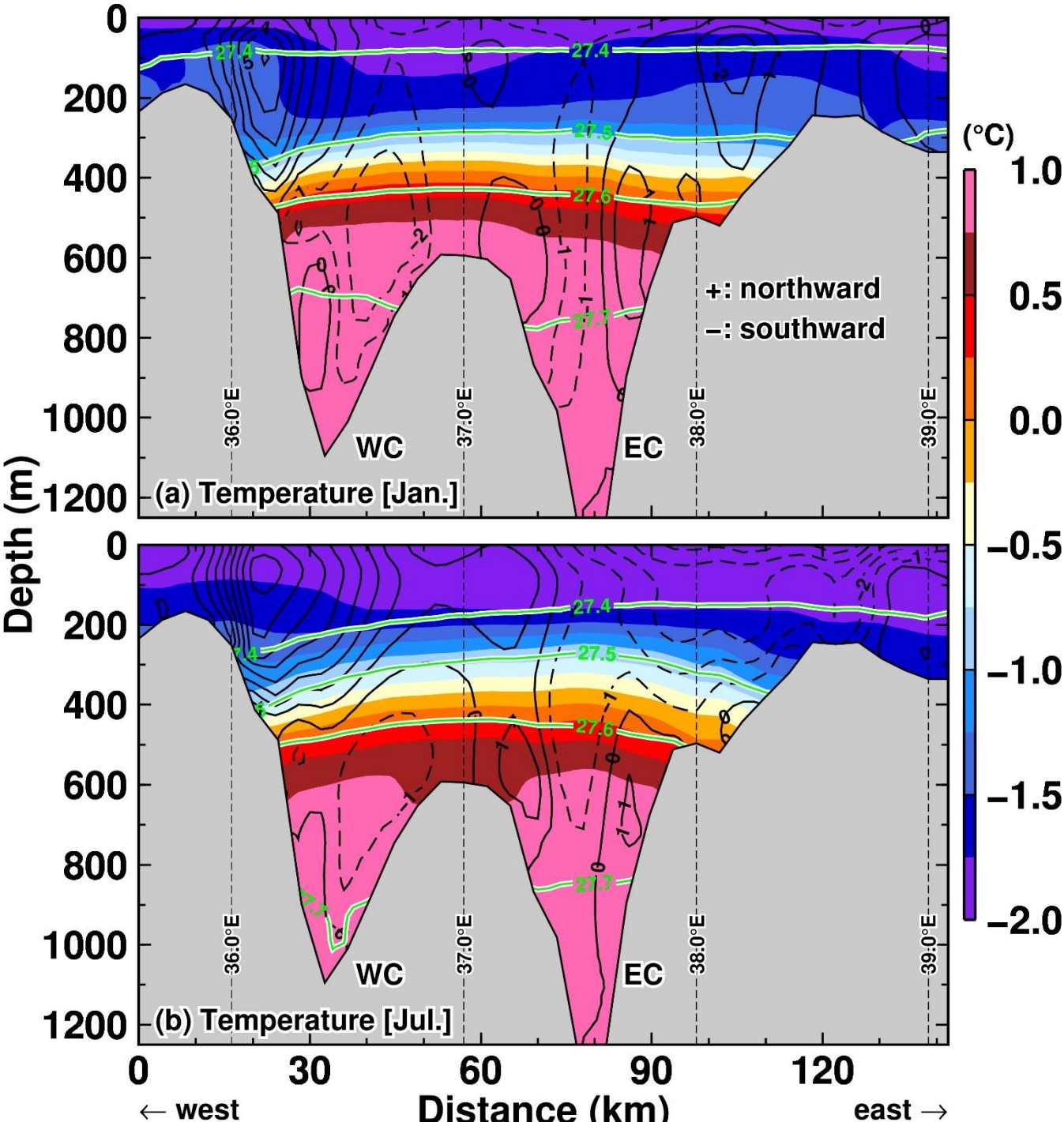

**Figure 15:** Vertical profiles of potential temperature along the sill section (at 68.5°S) in (a) January and (b) July in the NOFI case, with the velocity normal to the section (cm s$^{-1}$). Positive velocity indicates northward flow. Green curves indicate the contours of potential density anomalies. The model's variables were averaged over the period 2008–2018. The horizontal axis indicates the distance from the western side starting point of the sill section (see the line of the sill section in Fig. 8).

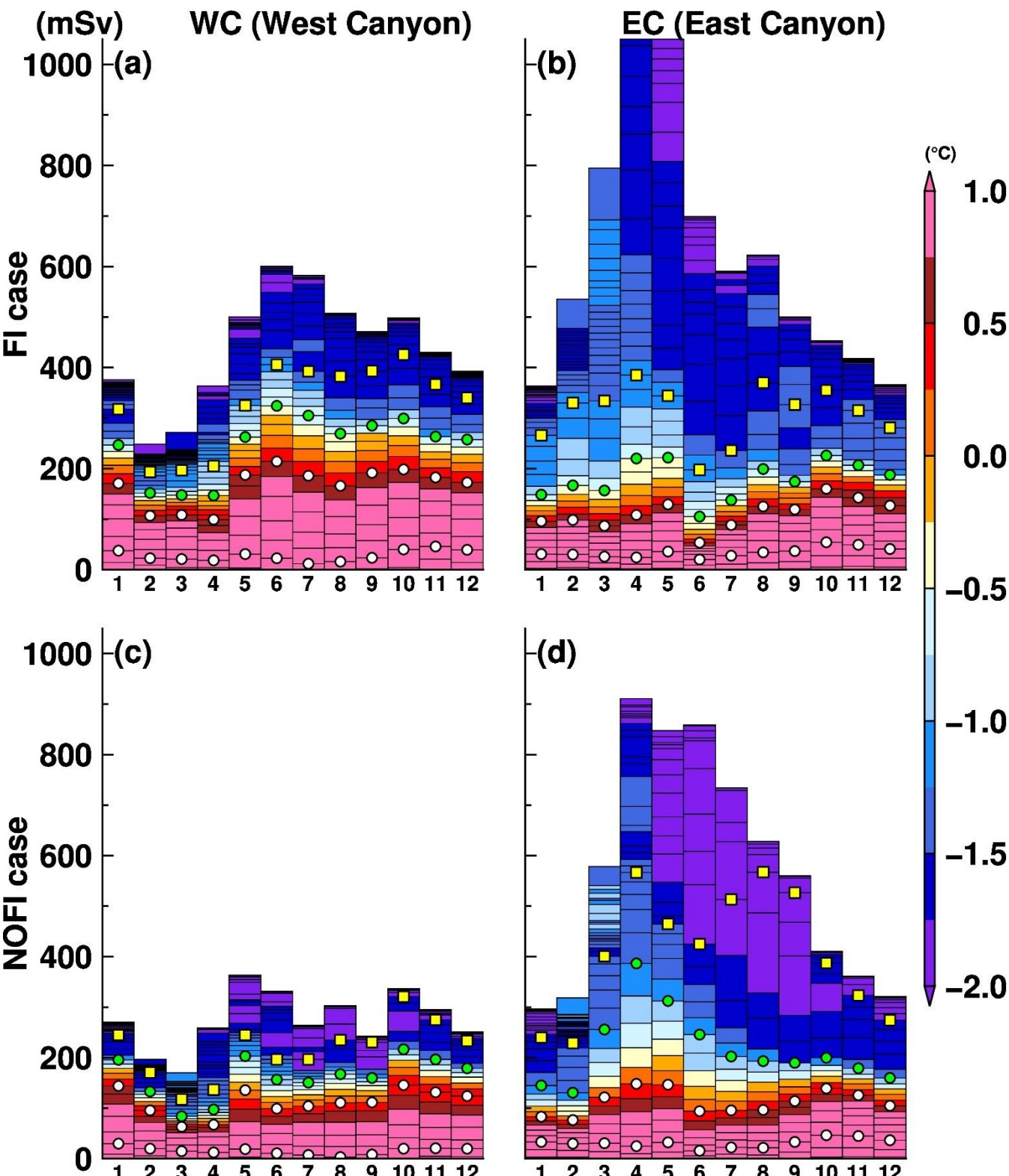

**Figure 16:** Seasonal variation of water mass transport flowing into the LHB through the two submarine canyons (left panels for WC and right panels for EC) with the potential temperature in potential density anomaly bin with 0.02 kg m$^{-3}$ (upper panels for the FI case and lower panels for the NOFI case). Green dots and yellow squares represent boundaries of 27.5 kg m$^{-3}$ and 27.4 kg m$^{-3}$ of the potential density anomaly, respectively. White does represent two density surfaces of 27.6 kg m$^{-3}$ and 27.7 kg m$^{-3}$. The variables were averaged over the period 2008–2018.

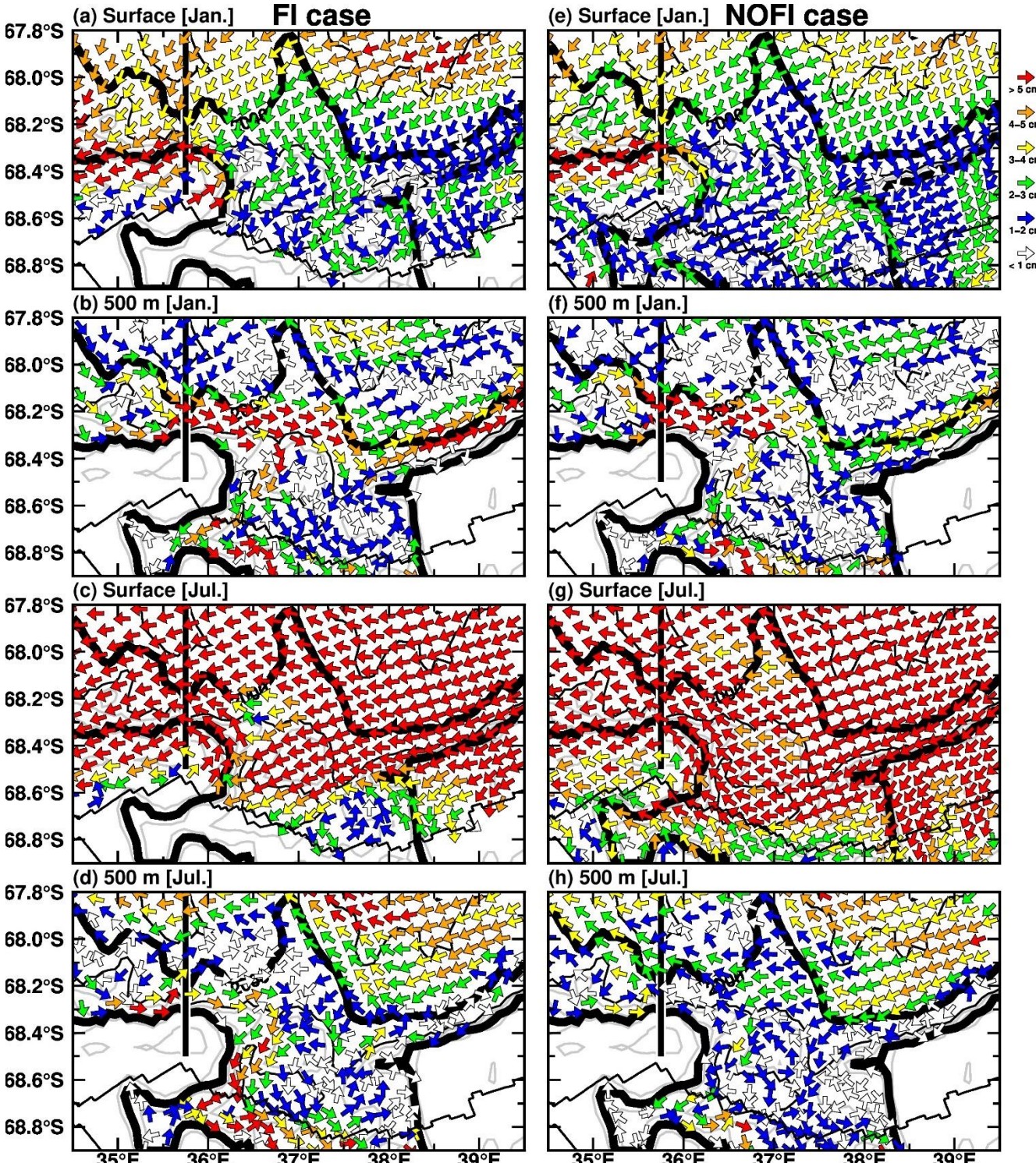

**Figure 17:** Maps of surface and intermediate-depth (at 500 m) ocean flows in January (upper four panels) and July (lower four panels). The vector shows the ocean flow direction, and the color shows the magnitude. Left and right panels show results from the FI and NOFI cases, respectively. The vertical black line indicates the sections used for Figs. 18 and 19. The black contours show the water depth with the 1000-m interval. The thick black curves show the 500-m and 2000-m depth contours to represent the upper continental slope. The zigzag lines are the coastline and the fast ice edge.

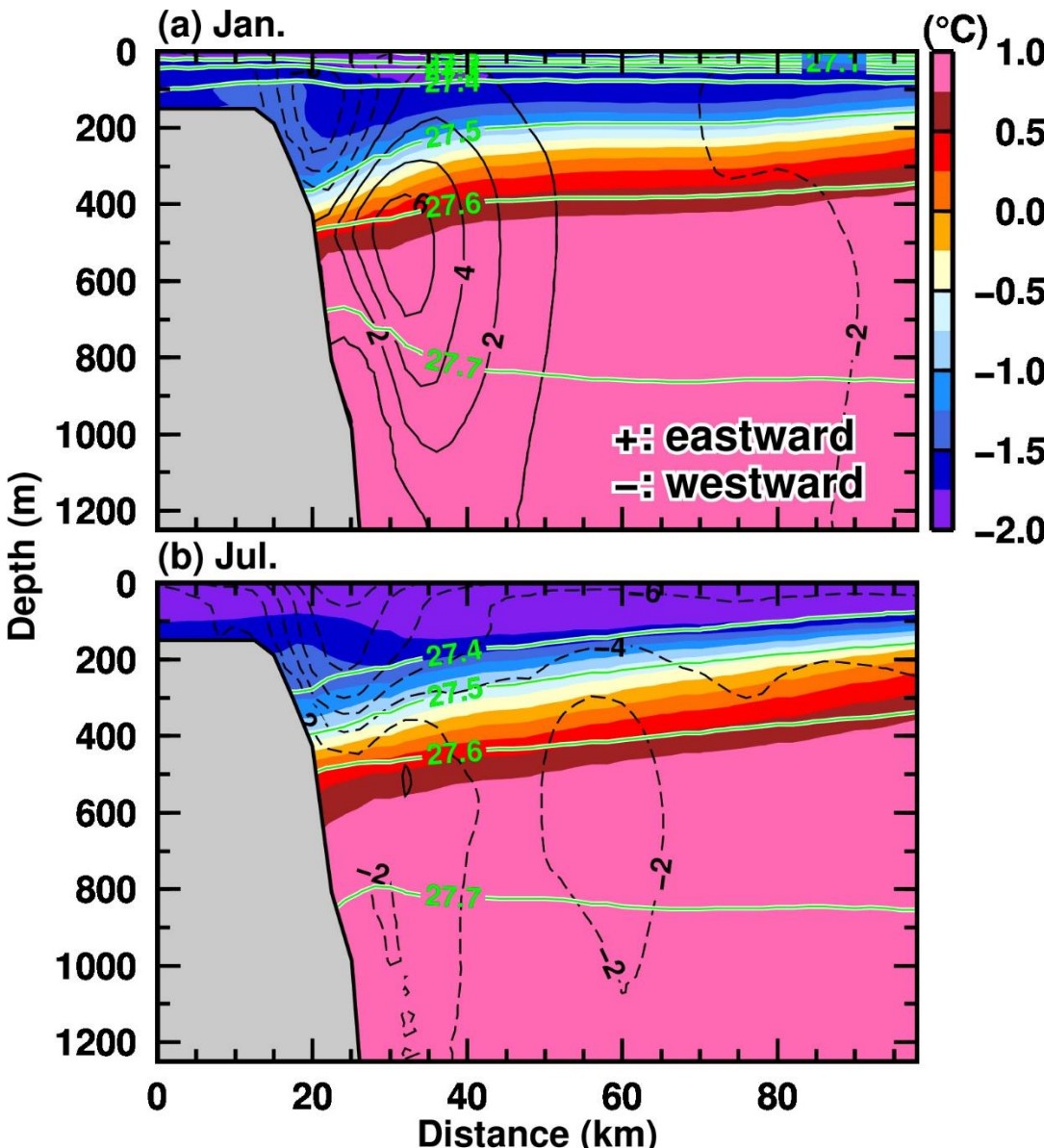

**Figure 18:** Vertical profiles of potential temperature (color), potential density anomaly (green contours), and east-westward velocity (black contours) along the north-south section at 35.75°E (see the black line in Fig. 17) in (a) January and (b) July in the NOFI case. Positive and negative values in the velocity indicate eastward and westward, respectively. The variables were averaged over the period 2008–2018.

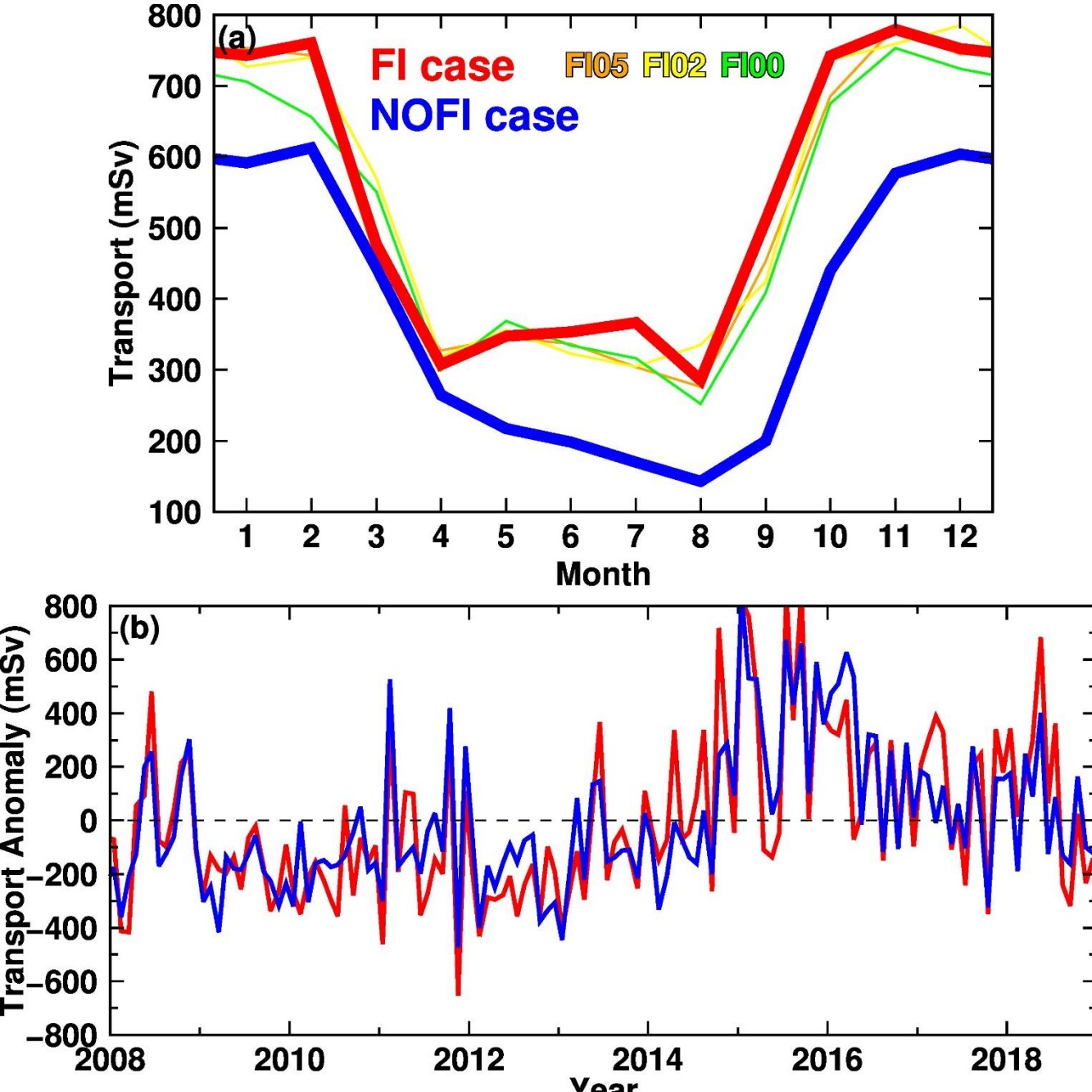

925 **Figure 19:** (a) Seasonal and (b) interannual variations of the eastward flow transport in the 200–800m depth across the section at 35.75°E (see the black line in Fig. 17). The monthly transport in panel (a) is climatology averaged over the period 2008–2018. Thin orange, yellow, and green lines are the results from the FIMOD series. In panel (b), the interannual variation is shown by the anomaly from the monthly climatology.

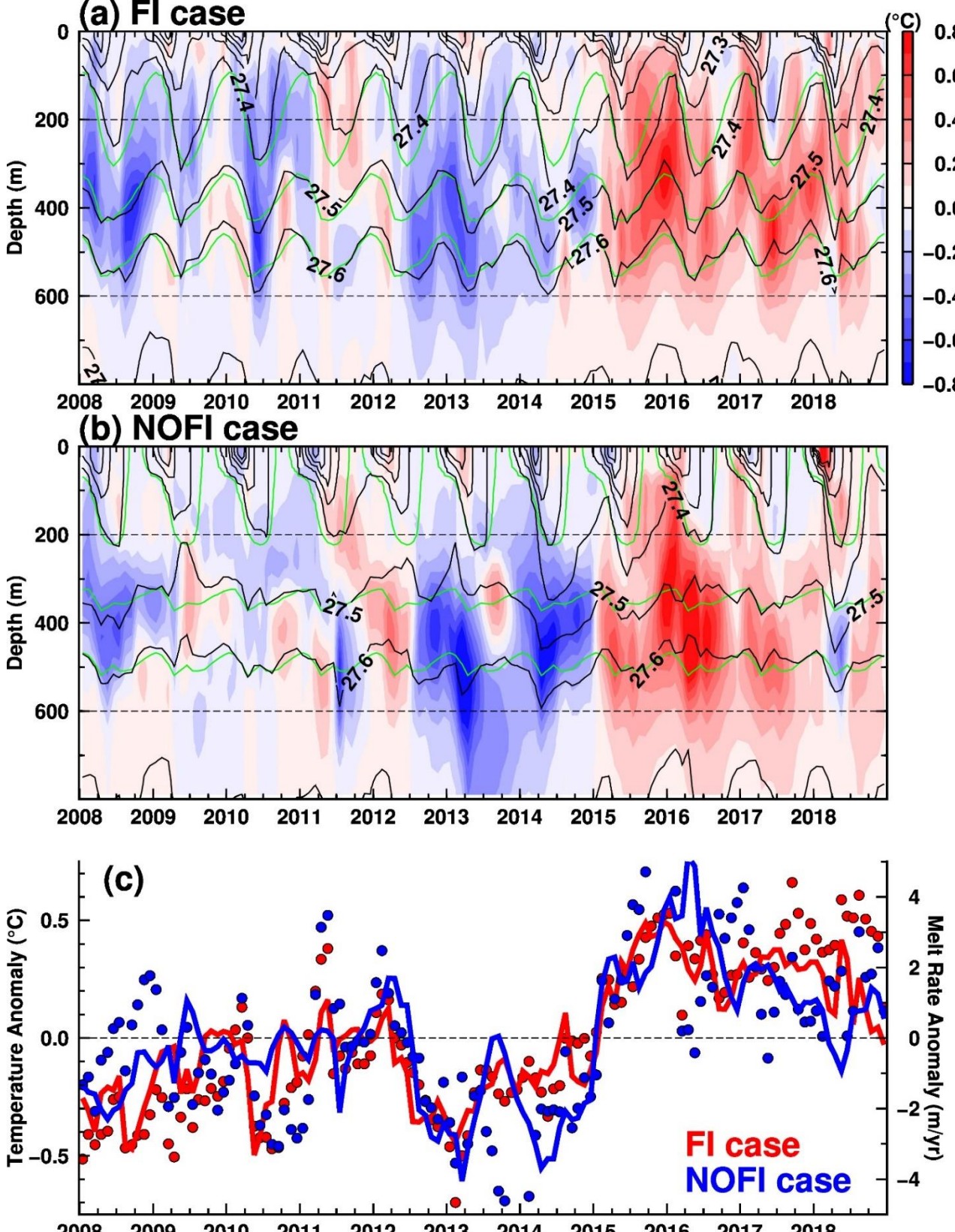

**Figure 20:** Time series of ocean temperature anomaly and potential density averaged over the box area in Fig. 8 (a: vertical profile in the FI case, b: vertical profile in the NOFI case, and c: 200–600m averaged temperature) and the SGT melt rate anomaly (dots in the panel c). The anomalies are deviations from the monthly climatologies averaged over the period 2008–2018. Green curves in panels a and b are climatologies of the potential density.