# Peer review of "Modeling intensive ocean-cryosphere interactions in Lützow-Holm Bay, East Antarctica"

_The Cryosphere, 2020_

## Referee Comment (RC1) · Anonymous Referee #1 · 25 Sep 2020

Kusahara and co-authors use an ocean-sea ice-ice shelf model to describe oceanographic and cryospheric processes in Lützow-Holm Bay, East Antarctica. Consistent with observations, the model shows warm intrusions onto the continental shelf and consequent rapid melting of the Shirase Glacier Tongue. The model also describes the impact of winds and fast ice on warm water delivery to the ice shelf. This study is an important advance to our understanding of the Antarctic system and therefore appropriate for The Cryosphere. However, I believe that some further explanation/analyses are required before the manuscript can be published, as described below.

Major Comments • A recent study by Hirano et al (2020) shows new observations from Lützow-Holm Bay. This new study by Kusahara et al represents an important advance compared to that previous study. However, in the text it is not fully clear what

[Figure]

is new here. I would try to strengthen what is new here (e.g. cross shelf exchange) and put less emphasis on what both model and observations show. • The simulation with fast ice shows warm water (CDW?) all over the water column in the trough. This is not observed anywhere around Antarctica. It seems to me that the no-fast-ice simulation produces more realistic results. A discussion about this is required, including whether the authors think the fast-ice simulation is realistic. • Figure 18: I like the schematic. However, how the water intrudes into the canyon is not analysed here. In particular, an eastward undercurrent requires cross-isobath flow (and potentially upwelling) to access the canyon on the eastern side. There is a wide literature describing this. The way this is depicted in the schematic is a bit misleading. I would closely look at the circulation pattern at the mouth of the canyon to see what happens.

Minor Comments

• Line 17: It is not well known if there is a gyre in East Antarctica, and specifically close to the Totten region. I would be more cautious with this inference. I would rather say that the ACC is closer to the coast in Amundsen/Bellingshausen seas and Totten region than elsewhere. • Line 49-54: see above. • Line 58: fastest melting? • Line 109: Concentration? What do you mean? • Line 186: "trough". • Line 189-190: change the wording to be more clear. • Figure 10: specify in the caption what the black lines are. • Line 290-291: "The observed estimate from the single location includes the entire SGT variability and regional variability." What do you mean? Please clarify. I would put in a map the location of the ApRES. I would also compare model basal melt in a grid point located near the ApRES location. • Line 330: "attached to the bottom". • Line 371: explained ->balanced. • Figure 17: specify in the caption the location of these data.

---

## Referee Comment (RC2) · Anonymous Referee #2 · 28 Sep 2020

In the manuscript "Modeling intensive ocean-cryosphere interactions in Luetzow-Holm Bay, East Antarctica", Kusahara et al. present results from a detailed, high-resolution model study of the sea ice-ice shelf-ocean system of Luetzow Holm Bay and the Shirase Glacier Tongue. The study examines the effect of fast ice on the flow of warm CDW towards the adjacent ice shelves through two sensitivity studies and the seasonal and interannual variability of the flow structure in Luetzow-Holm-Bay and at the shelf break.

I agree with the editor that there is sufficient novel material in the study with respect to Hirano et al. (2020) to recommend publication. Some of the figures are very similar to those in Hirano et al. (2020) and its supplementary material, but I will leave it to the editor to decide if they are sufficiently different.

[Figure]

I recommend publication of this study, but after major revisions. As I state below, I have my doubts whether using the ice shelf model unchanged to simulate the fast ice cover is appropriate. I would also like to see a more balanced, quantitative and detailed discussion of the drivers of the changes seen between the sensitivity runs and the interannual variability.

General comments

The authors base their model evaluation and analysis on two quasi-steady state runs with and without fast ice. For analysing the changes in the simulated circulation pattern and water mass distribution this is absolutely fine. However, the ocean observations they use to evaluate the model were taken in a period when there should still have been a transient response from the fast ice outbreak, if I understand the timing correctly (fast ice outbreak Mar-Apr 2016, ocean observations Jan 2017, ApRES Feb 2018-Jan 2019). While Fig. 4b indicates that the ApRES observations represent the NoFI case, I would like to see evidence how fast the modelled water column transitions to the NOFI case – something like a time series of ocean heat content or Hovmøller diagram of the temperature profile at a point in the bay would work. I do realise that the noFI case matches the CTD observations much better, but I would like to see that one winter is enough to change the water column to that state.

Your model treats the fast ice as ice shelf. The issue is that the computation of heat and freshwater fluxes in your ice shelf model uses exchange coefficients for heat and freshwater that imply an ocean surface speed of 15 cm/s, rather using the actual surface velocity as in the sea ice model. In the original Hellmer and Olbers (1989) model and follow ups this was adopted to simulate the effect of tides. Even though it is unclear which run this comes from and it is a long-term mean, your Fig. 12 makes me wonder if surface velocities are consistently (or ever) that large under the fast ice. Please check instantaneous surface speeds at a few points under the fast ice to confirm the actual magnitude of the surface speeds under the fast ice. My suspicion is that the freshwater input under fast ice is strongly overestimated due to the issue in the model formulation

that I have outlined above. You are clearly aware that the freshwater input in the FI case is large judging your discussion in Section 4, where you try to justify the magnitude of that freshwater input. However, in my opinion this model issue overestimates the difference between the FI and NOFI sensitivity runs.

There needs to be an earlier and stronger focus on the effect of surface buoyancy forcing and sea ice formation in the study. The change you make between the FI and NOFI sensitivity runs strongly affects the surface buoyancy forcing, but you largely ignore this effect up until late in the manuscript in favour of the changes in Ekman downwelling due to the changing winds. Section 4 needs to be moved to an earlier position in the paper. Section 5 really needs some time series of the forcing i.e. wind stress, Ekman pumping, and surface heat fluxes. Please add these as suitable time series. Considering how much emphasis you place on the Ekman pumping mechanism, a more quantitative discussion of this mechanism by showing Ekman pumping velocities or estimating their magnitude at the sea ice edge would strengthen your argument.

I spent a lot of time looking at Fig. 12, because I think the seasonal current reversal at the shelf break and inflow shift is fascinating and a great demonstration how the strength of the easterlies determines the flow direction at the shelf break. The stratification and tilt of the isopycnals towards the shelf below 100m doesn't change very much between January and July in your Fig 13. So in principle, all that happens is that changing winds shift that baroclinic shear profile back and forth between westward surface and eastward undercurrent in summer (Amundsen) and strong westward surface flow and weaker westward flow at the shelf break in winter (Weddell). What I am missing in the text is how this shifts the inflow between the two troughs from flowing predominantly through the western trough and outflow in the eastern trough in summer to a weaker inflow in the west and some inflow in the east in winter.

Specific points

l. 33-37: There is some duplication in these sentences that could be tightened.

l. 44-45: Amundsen Sea bottom topography isn't that certain either.

l. 49-54: I agree that Luetzow-Holm Bay is in a location equivalent to parts of the Amundsen Sea with respect to the Weddell and Ross Gyres being located south of their eastern flanks. However, in the Bellingshausen Sea the ACC flows along the shelf break and it is therefore distinctly different from the Amundsen. Also, both the Amundsen and Bellinghausen Seas have wide continental shelves that have their own circulation regimes, so to talk about the coast in that context is misleading. There are more appropriate publications to cite here: Ryan et al. (Deep Sea Research, 2016) investigated the eastern end of the Weddell, Dotto et al. (GRL, 2018) for Ross Gyre, and Armitage et al. (JGR 2018) identify the gyre circulation patterns over the entire Southern Ocean from satellite SSH observations.

l. 66-67: How do the 1990-92 observations compare to your FI case? How does the thickness of your simulated WW layer in the FI case compare to this?

l. 131: The reference for that new bathymetry is Hirano et al. (2020). The abstract states that it is "newly compiled" which would suggest to me that you are the first to present this bathymetry data set.

l. 132: 2.2 not 2.3. I really like the comparison of bathymetry products. Something that definitely needs more attention when modelling the circulation on Antarctic shelves.

l. 146-150: That hydrographic data set is old and Arctic-focussed. I am just curious why you chose it and not more recent data sets like WOA. However, since it is mainly used as initialization, I don't think it has a strong bearing on you model results. l. 146-150: What convection scheme do you use? Have you evaluated its suitability for the Southern Ocean?

l. 186: typo "trough" not "tough"

l. 186-187: Trough T4 is not labelled in Fig 2a, but in Fig. 6.

l. 190-191: But your new bathymetry is based on the JARE measurements. . .? Thus

using that data set to evaluate your model bathymetry seems inappropriate. Section 2.3 and Fig.3: I like the figure and presentation of the seasonal cycle, but I think you should show a mean over the years that go into your FI and NOFI cases, i.e., 2008-2018. Possibly also 2005-18.

l. 207-209: Already stated in the introduction, superfluous here. Maybe re-state later?

l. 218: should be Fig. 4b, since this shows the drift in the control run, the interannual variability and the two sensitivity runs. Figure 5 has only one panel.

l. 222-223: Please re-write this sentence.

l. 226-228: Please re-order your figures according to the order in the text. Since Fig. 5 is not used before Fig. 6, it needs to be moved. In addition, this sentence duplicates the figure caption to Fig. 6. SGT and SkG should be explained in the Fig.6 caption.

l. 233-241 & Fig. 7: You comment on the outer stations, but not E1 and A3 which show the largest discrepancies in salinity. The observations have S > 34.4 up to 400 m, i.e. there is a 400 m layer of mCDW in the observations that your model section does not show. Since this is the NOFI case I think this is being mixed away over the 13 years since you removed the fast ice. This also feeds back to my question after the convection scheme used and whether it is appropriate and whether the quasi-steady state NOFI case mean should be used to evaluate the model against observations. There is also the possibility that in the real ocean there wasn't enough convection in the first winter after the fast ice outbreak to erode the mCDW in front of SGT.

l. 243-250 and Fig. 8: After my previous comment, please add some evaluation of the simulated salinities as well to check how your simulated density structure compares and what the implications for the circulation might be.

l. 251-258: This is a great identification of the seasonal shift in inflow location, but his gets lost later?

l. 290-291: "The observed estimate from the single location includes the entire SGT

variability and regional variability." Not sure what you mean here. The next sentence states correctly that it is unclear how representative the ApRES location is. Please re-phrase or remove. N.B. Comparing the melt rate in your Fig. 6 with the location of the ApRES in Hirano et al. 2020, it appears that the instrument was located an area of the ice shelf with large gradients in melt rate. Thus it may be a tricky location for comparison. Have you done a point comparison to see if the model melt rate pattern shifts throughout the year?

l. 297-299: Your NOFI and FI sensitivity studies are designed to demonstrate that the seasonal cycle and upper ocean processes control the access of the warm water to the cavity. That is what Fig. 6-10 show: the model does have warm water on the shelf, but the surface processes control how much of it reaches the cavity. Remove or re-phrase this sentence.

l. 305-309 and Fig. 11: The seasonal cycle of the wind and of surface buoyancy forcing due to cooling and the addition of salt during sea ice formation coincide and using a model gives you the opportunity to pick these effects apart in more detail. The NOFI case shows strong evidence of sea ice formation causing first cooling and salinification of the surface and deep convection down to 350m around September. That seasonal cycle due to surface buoyancy forcing, partly due to a more strongly stratified upper ocean, is a lot weaker in the FI case (see general comments) and therefore you see the depression of the isopycnals due to Ekman downwelling more clearly.

Fig. 12 & 13: Unclear which run you show here.

Fig. 13 & 14: Please be consistent in defining flow directions as positive or negative. In Fig 13 eastward is negative, in Fig. 14 it is positive.

---

## Author Comment (AC1) · 19 Dec 2020

***Response to Reviewer #1's comments***
(NB *italicized text in box* is comments from the reviewer. Some numbers with bold fonts were inserted for our convenience in addressing the comments.)

> **(General comments)** *Kusahara and co-authors use an ocean-sea ice-ice shelf model to describe oceanographic and cryospheric processes in Lützow-Holm Bay, East Antarctica. Consistent with observations, the model shows warm intrusions onto the continental shelf and consequent rapid melting of the Shirase Glacier Tongue. The model also describes the impact of winds and fast ice on warm water delivery to the ice shelf. This study is an important advance to our understanding of the Antarctic system and therefore appropriate for The Cryosphere. However, I believe that some further explanation/analyses are required before the manuscript can be published, as described below.*
>
> *Major Comments*
> **(1)** *A recent study by Hirano et al (2020) shows new observations from Lützow-Holm Bay. This new study by Kusahara et al represents an important advance compared to that previous study. However, in the text it is not fully clear what is new here. I would try to strengthen what is new here (e.g. cross shelf exchange) and put less emphasis on what both model and observations show.*

**General comment and (1)**: Thank you very much for your careful reading, instructive suggestions, and understanding of this paper's importance. Following your advice, we have added detailed analyses and explanations for water exchange across the shelf breaks (Figs. 16 and 17). In this revision, we have changed the presentation of the model-observation comparison, taking account of both the reviewers' comments. With these modifications, we consider that manuscript's readability is much improved.

  We have highlighted the modifications with red color in the manuscript and quoted them in this response letter. In our response, we use new figure numbers in the revised manuscript to point out the corresponding figures. Since we have largely modified the manuscript, the figure numbers are different from the original manuscript.

  In the revision process, we found a mistake in the calculation of the eastward transport of the undercurrent (Fig. 19 in the revised manuscript). We have corrected the figure and correlation coefficients in Section 5, but these corrections don't change our conclusions.

> **(2)** *The simulation with fast ice shows warm water (CDW?) all over the water column in the trough. This is not observed anywhere around Antarctica. It seems to me that the no-fast-ice simulation produces more realistic results. A discussion about this is required, including whether the authors think the fast-ice simulation is realistic.*
> **(3)** *Figure 18: I like the schematic. However, how the water intrudes into the canyon is not analysed here. In particular, an eastward undercurrent requires cross-isobath flow (and potentially upwelling) to access the canyon on the eastern side. There is a wide literature describing this. The way this is depicted in the schematic is a bit misleading. I would closely look at the circulation pattern at the mouth of the canyon to see what happens.*

**(2)** In the revised manuscript, we have displayed vertical profiles of both ocean temperature and salinity along the TROUGH section in the FI and NOFI cases (Fig. 9) to see the CDW signals (characterized by warm and

high salinity). The intrusion of warm CDW along the ocean bottom was less modified when fast ice is present in the bay (FI case). In contrast, the ocean temperature at the surface-subsurface layer in the absence of the fast ice is substantially decreased due to atmosphere-ocean interaction in the bay (NOFI case). As you pointed out, we used the results from the NOFI case for several figures (Figs. 11, 15, and 18). We have added sentences to explain why we used the NOFI case for the analyses.

**L336–339**

In the end of this subsection, we compare the ocean temperature profile along the A-line between the observation and NOFI case (Fig. 11). As shown above, the observations seem to capture the transient conditions. Since the observed temperature profile along the TROUGH section is more similar to that in the NOFI case (Fig. 9), here we use the results from the NOFI case for the comparison along the A-line.

**L456–458**

In order to examine the detailed characteristics of the undercurrent on the upper continental slope, vertical profiles of ocean temperature, density, and east-westward flow along a section at 35.75°E (see the black line in Fig. 17) are plotted in Figure 18 for the NOFI case, which shows a clear seasonality in the undercurrent.

(3) Thank you for your comment. We have added the detailed ocean flow around the sill region (Fig. 17 for the FI and NOFI cases) and seasonal variation of the inflow transport to the bay across the sill section (Fig.16 for the FI and NOFI cases). These figures clearly illustrate the seasonality in the warm inflow into the bay. In the revision, we removed the schematic figure to reduce the number of figures. The detailed ocean flows in both cases (Fig. 17) seem to be better for this paper than the simplistic schematic figure.
* * *
*Minor Comments*

**(4)** *Line 17: It is not well known if there is a gyre in East Antarctica, and specifically close to the Totten region. I would be more cautious with this inference. I would rather say that the ACC is closer to the coast in Amundsen/Bellingshausen seas and Totten region than elsewhere.*

**(5)** *Line 49-54: see above.*

**(6)** *Line 58: fastest melting?*

**(7)** *Line 109: Concentration? What do you mean?*

**(8)** *Line 186: "trough".*

**(9)** *Line 189- 190: change the wording to be more clear.*

**(10)** *Figure 10: specify in the caption what the black lines are.*
* * *
**(4)(5)** We have rephrased them.

**L23–24**

Both regions have a common oceanographic feature: southward deflection of the Antarctic Circumpolar Current brings CDW onto the continental shelves.

**L54–63 ### (adding several references)**

Ice shelves in the Amundsen and Bellingshausen Seas and the Totten ice shelves have a common feature: the Antarctic Circumpolar Current (ACC) is proximal to the Antarctic continental shelf regions, and thus the CDW can affect regional coastal water masses. The Southern Ocean has three large-scale cyclonic (subpolar) gyres: Ross, Kerguelen, and Weddell Gyres (Gordon, 2008). Amundsen and Bellingshausen Seas and the offshore

region of Wilkes Land correspond to south-eastern sides of the Ross and Kerguelen Gyres, respectively, where the southward deflection of the large-scale ocean circulations brings warm water poleward to the Antarctic continental shelves (Armitage et al., 2018; Dotto et al., 2018; Gille et al., 2016; McCartney and Donohue, 2007; Mizobata et al., 2020). By analogy to the relationship between the southward deflection of the ocean circulations and the active ice shelf melting areas, one can naturally speculate active ocean-ice shelf interaction around the south-eastern side of the Weddell Gyre (Ryan et al., 2016).

**(6)** We have rephrased the sentence.
**L64–65**
This glacier, Shirase Glacier in Lützow-Holm Bay (LHB), Enderby Land, East Antarctica, is one of the fastest laterally-flowing glaciers around Antarctica (Nakamura et al., 2010; Rignot, 2002).

**(7)** We have rephrased the sentence.
**L115–117**
Prognostic equations for the sea-ice momentum, mass, and concentration (ice-covered area) were taken from Mellor and Kantha (1989). Internal ice stress was formulated by the elastic-viscous-plastic rheology (Hunke and Dukowicz, 1997), and sea ice salinity was fixed at 5 psu.

**(8)** We have corrected it (L208).

**(9)** We have rephrased the sentence.
**L209–212**
In the southern part of the trough, there is a north-southward elongated depression at the latitudes from 69.5°S to 70°S with a maximum depth of approximately 1600 m. The feature is confirmed in the observational data. We consider that our dataset is the best estimate of bottom topography in the LHB region at this moment in which available bathymetric observations were taken into account.

**(10)** We have corrected the caption (Fig. 14).

> **(11)** *Line 290-291: "The observed estimate from the single location includes the entire SGT variability and regional variability." What do you mean? Please clarify. I would put in a map the location of the ApRES. I would also compare model basal melt in a grid point located near the ApRES location.*
> **(12)** *Line 330: "attached to the bottom".*
> **(13)** *Line 371: explained ->balanced.*
> **(14)** *Figure 17: specify in the caption the location of these data.*

**(11)** Thank you for your suggestions. We have rephrased the sentences and added a comparison of local basal melt rate at around the ApRES location (thin lines in Fig. 7).
**L405–412**
These general agreements provide some confidence for the model results in this study, especially for the seasonality of the SGT basal melt rate. It should be noted that the ApRES located on the high-gradient zone

of the basal melt rate in the model (Fig. 8) and the observation period was less than one year. When comparing the local basal melt rate between the model and observation, the model underestimates the seasonal amplitude of the basal melt rate. This indicates the model underestimates the seasonal cycle of the warm water inflow to the cavity under the ApRES position. The bottom topography under the SGT is outside of our compiled topography data, and thus it was not well constrained by the observed topography. The uncertainty of the bottom topography probably leads to the discrepancy in the seasonal amplitude of the local basal melt rate.

**(12) (13)** We have corrected them (L462 and L257).

**(14)** We have corrected the caption for Fig. 20. ("Time series of ocean temperature anomaly and potential density averaged over the box area in Fig. 8 …")

---

## Author Comment (AC2) · 19 Dec 2020

**Response to Reviewer #2's comments**

(NB *italicized text in box* is comments from the reviewer. Some numbers with bold fonts were inserted for our convenience in addressing the comments.)

> *In the manuscript "Modeling intensive ocean-cryosphere interactions in Luetzow-Holm Bay, East Antarctica", Kusahara et al. present results from a detailed, high-resolution model study of the sea ice-ice shelf-ocean system of Luetzow Holm Bay and the Shirase Glacier Tongue. The study examines the effect of fast ice on the flow of warm CDW towards the adjacent ice shelves through two sensitivity studies and the seasonal and interannual variability of the flow structure in Luetzow-Holm-Bay and at the shelf break.*
>
> *I agree with the editor that there is sufficient novel material in the study with respect to Hirano et al. (2020) to recommend publication. Some of the figures are very similar to those in Hirano et al. (2020) and its supplementary material, but I will leave it to the editor to decide if they are sufficiently different.*
>
> *I recommend publication of this study, but after major revisions. As I state below, I have my doubts whether using the ice shelf model unchanged to simulate the fast ice cover is appropriate. I would also like to see a more balanced, quantitative and detailed discussion of the drivers of the changes seen between the sensitivity runs and the interannual variability.*

Thank you very much for your careful reading, instructive suggestions, and understanding of this paper's importance. This is a complementary paper to Hirano et al. (2020), paying more attention to the ice-ocean interaction in LHB from a modeling perspective. As you pointed out, we used almost the same figures (not exactly the same due to the averaging period) in the two papers, but we consider that sharing the key figures would be a nice bridge between the two papers.

In this revision, we have taken into account your comments. We have substantially modified the manuscript by adding new analyses/experiments/figures and changing the paper's structure. We appreciated your kind comments for enhancing the quality of this study.

We have highlighted the modifications with red color in the manuscript and quoted them in this response letter. In our response, we use new figure numbers in the revised manuscript to point out the corresponding figures. Since we have largely modified the manuscript, the figure numbers are different from the original manuscript.

In the revision process, we found a mistake in calculating the eastward transport of the undercurrent (Fig. 19 in the revised manuscript). We have corrected the figure and correlation coefficients in Section 5, but these corrections don't change our conclusions.

*General comments*

**(1)** *The authors base their model evaluation and analysis on two quasi-steady state runs with and without fast ice. For analysing the changes in the simulated circulation pattern and water mass distribution this is absolutely fine. However, the ocean observations they use to evaluate the model were taken in a period when there should still have been a transient response from the fast ice outbreak, if I understand the timing correctly (fast ice outbreak Mar-Apr 2016, ocean observations Jan 2017, ApRES Feb 2018-Jan 2019). While Fig. 4b indicates that the ApRES observations represent the NoFI case, I would like to see evidence how fast the modelled water column transitions to the NOFI case – something like a time series of ocean heat content or Hovmøller diagram of the temperature profile at a point in the bay would work. I do realise that the noFI casematches the CTD observations much better, but I would like to see that one winter is enough to change the water column to that state.*

**(2)** *Your model treats the fast ice as ice shelf. The issue is that the computation of heat and freshwater fluxes in your ice shelf model uses exchange coefficients for heat and freshwater that imply an ocean surface speed of 15 cm/s, rather using the actual surface velocity as in the sea ice model. In the original Hellmer and Olbers (1989) model and follow ups this was adopted to simulate the effect of tides. Even though it is unclear which run this comes from and it is a long-term mean, your Fig. 12 makes me wonder if surface velocities are consistently (or ever) that large under the fast ice. Please check instantaneous surface speeds at a few points under the fast ice to confirm the actual magnitude of the surface speeds under the fast ice. My suspicion is that the freshwater input under fast ice is strongly overestimated due to the issue in the model formulation that I have outlined above. You are clearly aware that the freshwater input in the FI case is large judging your discussion in Section 4, where you try to justify the magnitude of that freshwater input. However, in my opinion this model issue overestimates the difference between the FI and NOFI sensitivity runs.*

**(1)** Thank you very much for your suggestion about the analysis of the transient response. In the revised manuscript, we display the vertical profiles of the ocean temperature and salinity along the TROUGH section in both the FI and NOFI cases (Fig. 9) to clearly show the differences between the two cases. Following your advice, we have added analysis and a figure (Fig. 10) showing the transition of coastal water masses and SGT-melting from the FI case to the NOFI case. The new results reveal that it takes a few years to adjust the coastal water masses and SGT melting (Please refer to Section 4.1 "Transient response and quasi-equilibrium of the CDW intrusion into LHB and the SGT melting").

**(2)** Your understanding is right. We have added additional experiments in which the exchange coefficients were reduced only under the fast ice (FIMOD series) and added a new table for basal melting at fast ice and the surrounding glaciers (Table 1). A comparison among the baseline experiments (the FI and NOFI cases) and the additional experiments demonstrated that meltwater from the fast ice has minor effects on our main results. We have added results from the additional experiments in several figures to see the impacts (Figs. 6 and 19). We have adequately added paragraphs to include the additional experiments and results (Last paragraph in Section 2.1 and last paragraph in Section 3). Thank you very much for your insightful comments.

> **(3-a)** *There needs to be an earlier and stronger focus on the effect of surface buoyancy forcing and sea ice formation in the study. The change you make between the FI and NOFI sensitivity runs strongly affects the surface buoyancy forcing, but you largely ignore this effect up until late in the manuscript in favour of the changes in Ekman downwelling due to the changing winds. Section 4 needs to be moved to an earlier position in the paper.* **(3-b)** *Section 5 really needs some time series of the forcing i.e. wind stress, Ekman pumping, and surface heat fluxes. Please add these as suitable time series. Considering how much emphasis you place on the Ekman pumping mechanism, a more quantitative discussion of this mechanism by showing Ekman pumping velocities or estimating their magnitude at the sea ice edge would strengthen your argument.*
>
> **(4)** *I spent a lot of time looking at Fig. 12, because I think the seasonal current reversal at the shelf break and inflow shift is fascinating and a great demonstration how the strength of the easterlies determines the flow direction at the shelf break. The stratification and tilt of the isopycnals towards the shelf below 100m doesn't change very much between January and July in your Fig 13. So in principle, all that happens is that changing winds shift that baroclinic shear profile back and forth between westward surface and eastward undercurrent in summer (Amundsen) and strong westward surface flow and weaker westward flow at the shelf break in winter (Weddell). What I am missing in the text is how this shifts the inflow between the two troughs from flowing predominantly through the western trough and outflow in the eastern trough in summer to a weaker inflow in the west and some inflow in the east in winter.*

**(3-a)** Following your suggestion, we have moved the content to the earlier part of the manuscript with a new Section 3 "Difference in the surface forcing in the experiments with and without fast ice cover" (L229–271). We consider this modification enhances the readability of this paper. Thank you very much for your advice.

**(3-b)** We have added results of the monthly Ekman downwelling (Fig. 13) and sea-ice production (Fig. 4). The comparison of the modeled ocean density surfaces with the monthly Ekman downwelling confirmed that the model could capture the wind-driven processes proposed in the previous studies (Ohshima et al. 1996 and Hirano et al. 2020).

**(4)** In this revision, we have added a seasonal variation of the inflow transport through the two submarine canyons in the sill section (Fig. 16), along with the ocean flow patterns in the FI and NOFI cases (Fig. 17). ### L439–454 ###

Next, we show the spatial distribution of ocean flows at the surface and 500 m depth to identify the origin of the warm water mass flowing to LHB through the submarine canyons (Fig. 17). South-westward flows along the continental slope predominate surface flows throughout the year. This alongshore surface flow is strong in winter and weak in summer (panels a, c, e, and g in Fig. 17), which corresponds to the seasonality of surface wind magnitude in this region (Fig. 3i). Ocean flow patterns at 500-m depth are quite different from the surface flow patterns. In summer, a strong eastward flow is produced on the continental slope between the 500-m and 2000-m isobaths. In this study, we use the term "undercurrent" to point out the eastward flow below the westward surface flow. Although the eastward undercurrent is found in both the FI and NOFI cases, the undercurrent in the FI case is more vigorous than in the NOFI case. A part of the undercurrent is redirected southward at the submarine canyons. The southward flow brings warm water in the deep layer into LHB (Fig. 17b, see also Figs. 15a and 12). The southward transports of the warm water through the two submarine

canyons are comparable to each other (Fig. 16). In the winter, the undercurrent weakens in the FI case and almost disappears in the NOFI case. In the FI case, the eastward flows are found only on the western side of the sill section. In the NOFI case and the eastern part of the sill section in the FI case, the direction of the subsurface flow on the continental slope becomes westward, which is the same direction of the surface flow and the wind. Also, in the winter, a part of the westward subsurface flow is redirected southward at the submarine canyons, bringing warm water into the inside of the bay (Figs. 17d and 17h, see also Fig. 15b). In the FI case, the weak eastward undercurrent also contributes to the southward flow across the WC (Fig. 17d).
* * *
*Specific points*

**(5)** *l. 33-37: There is some duplication in these sentences that could be tightened.*

**(6)** *l. 44-45: Amundsen Sea bottom topography isn't that certain either.*

**(7)** *l. 49-54: I agree that Luetzow-Holm Bay is in a location equivalent to parts of the Amundsen Sea with respect to the Weddell and Ross Gyres being located south of their eastern flanks. However, in the Bellingshausen Sea the ACC flows along the shelf break and it is therefore distinctly different from the Amundsen. Also, both the Amundsen and Bellinghausen Seas have wide continental shelves that have their own circulation regimes, so to talk about the coast in that context is misleading. There are more appropriate publications to cite here: Ryan et al. (Deep Sea Research, 2016) investigated the eastern end of the Weddell, Dotto et al. (GRL, 2018) for Ross Gyre, and Armitage et al. (JGR 2018) identify the gyre circulation patterns over the entire Southern Ocean from satellite SSH observations.*
* * *
**(5)** We have rephrased the sentence.

**L40–42**

Several observational and modeling studies have pointed out that enhanced intrusions of warm waters onto some Antarctic continental shelf regions trigger more active ocean-ice shelf interaction (i.e., ice-shelf basal melting), leading to the negative mass balance of the Antarctic ice sheet (Depoorter et al., 2013; Rignot et al., 2013).

**(6)** We have rephrased it. (L49 "the ice shelves in this sector have been relatively well studied")

**(7)** Thank you very much for your information about the papers for the paragraph. We have added references.

**L54–62**

… Ice shelves in the Amundsen and Bellingshausen Seas and the Totten ice shelves have a common feature: the Antarctic Circumpolar Current (ACC) is proximal to the Antarctic continental shelf regions, and thus the CDW can affect regional coastal water masses. The Southern Ocean has three large-scale cyclonic (subpolar) gyres: Ross, Kerguelen, and Weddell Gyres (Gordon, 2008). Amundsen and Bellingshausen Seas and the offshore region of Wilkes Land correspond to south-eastern sides of the Ross and Kerguelen Gyres, respectively, where the southward deflection of the large-scale ocean circulations brings warm water poleward to the Antarctic continental shelves (Armitage et al., 2018; Dotto et al., 2018; Gille et al., 2016; McCartney and Donohue, 2007; Mizobata et al., 2020). By analogy to the relationship between the southward deflection of the ocean circulations and the active ice shelf melting areas, one can naturally speculate active ocean-ice shelf interaction around the south-eastern side of the Weddell Gyre (Ryan et al., 2016). In fact, …

**(8)** *l. 66-67: How do the 1990-92 observations compare to your FI case? How does the thickness of your simulated WW layer in the FI case compare to this?*

**(9)** *l. 131: The reference for that new bathymetry is Hirano et al. (2020). The abstract states that it is "newly compiled" which would suggest to me that you are the first to present this bathymetry data set.*

**(10)** *l. 132: 2.2 not 2.3. I really like the comparison of bathymetry products. Something that definitely needs more attention when modelling the circulation on Antarctic shelves.*

**(11)** *l. 146-150: That hydrographic data set is old and Arctic-focussed. I am just curious why you chose it and not more recent data sets like WOA. However, since it is mainly used as initialization, I don't think it has a strong bearing on you model results.*

**(12)** *l. 146-150: What convection scheme do you use? Have you evaluated its suitability for the Southern Ocean?*

**(13)** *l. 186: typo "trough" not "tough"*

**(14)** *l. 186-187: Trough T4 is not labelled in Fig 2a, but in Fig. 6.*

**(8)** Thank you for your suggestion. Since the period of the model integration is from 2005 to 2018 and 90's observation is outside of our focal period, we didn't include the comparison with the data in this study.

**(9)** We have rephrased it throughout the manuscript with the expression "recently-compiled new dataset" (L27)/"recently-compiled" (L141).

**(10)** Thank you for your understanding of our research.

**(11)** It is a historical reason for our model developments. Our regional sea ice-ocean model with ice-shelf component is derived from a global sea ice-ocean model (COCO), which is a part of a climate model. We used PHC for our model, being consistent with the global OGCM.

   The horizontal resolution of the hydrographic dataset is much coarser than the model horizontal resolution of our focal regions. The ocean structures that we examined in this study were not resolved in the dataset. The modeled features discussed in this study were produced in the model. Of course, the hydrographic datasets' selection has some effects on the modeled fields, but we consider that the selection doesn't largely change our model results and conclusions.

**(12)** We have added sentences to briefly describe the mixing schemes in the model section.
**L117–120**
Unstable stratification produced in the ocean model was removed by a convective adjustment, and a surface mixed layer scheme (Noh and Kim, 1999) was used for the open ocean, as in our previous study of a circumpolar Southern Ocean modeling that showed a reasonable representation of seasonal mixed layer depth over the Southern Ocean (Kusahara et al., 2017).

**(13)** We have corrected it (L208)

**(14)** We have added labels of the troughs in the figure (Fig. 2a).

**(15-a)** *l. 190-191: But your new bathymetry is based on the JARE measurements. . .? Thus using that data set to evaluate your model bathymetry seems inappropriate.* **(15-b)** *Section 2.3 and Fig.3: I like the figure and presentation of the seasonal cycle, but I think you should show a mean over the years that go into your FI and NOFI cases, i.e., 2008-2018. Possibly also 2005-18.*

**(16)** *l. 207-209: Already stated in the introduction, superfluous here. Maybe re-state later?*

**(17)** *l. 218: should be Fig. 4b, since this shows the drift in the control run, the interannual variability and the two sensitivity runs. Figure 5 has only one panel.*

**(18)** *l. 222-223: Please re-write this sentence.*

**(19)** *l. 226-228: Please re-order your figures according to the order in the text. Since Fig. 5 is not used before Fig. 6, it needs to be moved. In addition, this sentence duplicates the figure caption to Fig. 6. SGT and SkG should be explained in the Fig.6 caption.*

**(20)** *l. 233-241 & Fig. 7: You comment on the outer stations, but not E1 and A3 which show the largest discrepancies in salinity. The observations have S > 34.4 up to 400 m, i.e. there is a 400 m layer of mCDW in the observations that your model section does not show. Since this is the NOFI case I think this is being mixed away over the 13 years since you removed the fast ice. This also feeds back to my question after the convection scheme used and whether it is appropriate and whether the quasi-steady state NOFI case mean should be used to evaluate the model against observations. There is also the possibility that in the real ocean there wasn't enough convection in the first winter after the fast ice outbreak to erode the mCDW in front of SGT.*

**(21)** *l. 243-250 and Fig. 8: After my previous comment, please add some evaluation of the simulated salinities as well to check how your simulated density structure compares and what the implications for the circulation might be.*

**(15-a)** The sentence was removed.

**(15-b)** Following your comments, we have changed the averaging period to 2008–2018.

**(16)** In the revised manuscript, the sentences were used for a short introduction in a new subsection "4.2 Seasonal changes in Ekman downwelling and density surfaces".

**(17)** We have corrected it. We have checked the consistency between the figure numbers and citations throughout the manuscript.

**(18) (19)** The corresponding sentences were removed or largely modified in the revision.

**(20)(21)** In the revision, we have added panels of the vertical salinity profiles in the FI case (Fig. 9). We have added comments on the salinity biases at the southernmost stations. As you suggested, the observation likely to capture the salinity transition from the fast ice breakup, and thus the observed salinity profiles at the southern stations are more similar to the FI case.

**L314–322**

There are some model biases in salinity. The observed salinity near the bottom throughout the section is higher by more than 0.1 psu than in the model. Since the less saline signal in the model is found near the sill region, the salinity biases probably come from an insufficient representation of salinity values on the continental slope

and rise regions. Even taking into account the salinity bias throughout the section, the near-bottom salinity at the southernmost stations (E1 and A3) is substantially underestimated in both cases, in particular in the NOFI case. The near-bottom salinity in the FI case is higher than in the NOFI case, and thus the FI case appears to better represent the near-bottom water salinity at the southernmost stations. The better agreement of the bottom salinity between the observation and FI case suggests that the near bottom signal responds more slowly to the fast ice breakup and that the in-situ observations within one year from the breakup event captured the transitions of the coastal water masses.
* * *
**(22)** *l. 251-258: This is a great identification of the seasonal shift in inflow location, but his gets lost later?*

**(23)** *l. 290-291: "The observed estimate from the single location includes the entire SGT variability and regional variability." Not sure what you mean here. The next sentence states correctly that it is unclear how representative the ApRES location is. Please re-phrase or remove. N.B. Comparing the melt rate in your Fig. 6 with the location of the ApRES in Hirano et al. 2020, it appears that the instrument was located an area of the ice shelf with large gradients in melt rate. Thus it may be a tricky location for comparison. Have you done a point comparison to see if the model melt rate pattern shifts throughout the year?*
* * *
**(22)** In this revision, we have added a figure to show the seasonal variation of inflow transport across the sill section through the two troughs (Fig 16). Using this figure, vertical profiles of temperature and north-southward velocity (Fig 15), and spatial distribution of surface and subsurface ocean flows (Fig. 17) shows a clear illustration of the seasonal inflow into LHB. (see subsection 4.4 "Warm water intrusions from the continental shelf break and slope regions to LHB" )

**(23)** We have rephrased the sentences and added a comparison of local basal melt rate at around the ApRES location.

**L405–412**

These general agreements provide some confidence for the model results in this study, especially for the seasonality of the SGT basal melt rate. It should be noted that the ApRES located on the high-gradient zone of the basal melt rate in the model (Fig. 8) and the observation period was less than one year. When comparing the local basal melt rate between the model and observation, the model underestimates the seasonal amplitude of the basal melt rate. This indicates the model underestimates the seasonal cycle of the warm water inflow to the cavity under the ApRES position. The bottom topography under the SGT is outside of our compiled topography data, and thus it was not well constrained by the observed topography. The uncertainty of the bottom topography probably leads to the discrepancy in the seasonal amplitude of the local basal melt rate.

**(24)** *l. 297-299: Your NOFI and FI sensitivity studies are designed to demonstrate that the seasonal cycle and upper ocean processes control the access of the warm water to the cavity. That is what Fig. 6-10 show: the model does have warm water on the shelf, but the surface processes control how much of it reaches the cavity. Remove or re-phrase this sentence.*

**(25)** *l. 305-309 and Fig. 11: The seasonal cycle of the wind and of surface buoyancy forcing due to cooling and the addition of salt during sea ice formation coincide and using a model gives you the opportunity to pick these effects apart in more detail. The NOFI case shows strong evidence of sea ice formation causing first cooling and salinification of the surface and deep convection down to 350m around September. That seasonal cycle due to surface buoyancy forcing, partly due to a more strongly stratified upper ocean, is a lot weaker in the FI case (see general comments) and therefore you see the depression of the isopycnals due to Ekman downwelling more clearly.*

**(26)** *Fig. 12 & 13: Unclear which run you show here.*

**(27)** *Fig. 13 & 14: Please be consistent in defining flow directions as positive or negative. In Fig 13 eastward is negative, in Fig. 14 it is positive.*

**(24)** We have rephrased the sentences.

**L422–425**

The previous analyses in this section demonstrated that the warm water intrusions onto the continental shelf predominate the ocean structure in LHB (Figs. 8, 9, and 11), the water masses flowing into the SGT cavity (Fig. 14), and the magnitude of the SGT basal melting (Figs. 6 and 7). Here, we examine in detail water mass exchanges across the sill section through the submarine canyons and ocean flow on the upper continental slope regions.

**(25)** In this revision, we have added seasonal and interannual variations of sea ice production in the NOFI case (Fig. 5) and the time evolution of the ocean properties' differences between the FI and NOFI cases. In the earlier part of the manuscript (Section 3 "Difference in the surface forcing in the experiments with and without fast ice cover"), we clearly discussed the surface forcing difference in the two baseline experiments.

**(26) (27)** We have corrected them.

---

## Author Comment (AC3) · 19 Dec 2020

The comment was uploaded in the form of a supplement:
https://tc.copernicus.org/preprints/tc-2020-240/tc-2020-240-AC3-supplement.pdf

―――――――――――――――

---

## Author Response (AR2)

Dear Editor Dr. Nicolas Jourdain,

Thank you very much for your thoughtful treatment of our article "Modeling intensive ocean–cryosphere interactions in Lützow-Holm Bay, East Antarctica" by Kusahara and others (tc-2020-240). We have read and addressed all of the comments/corrections from the reviewer. Please find the revised manuscript and our detailed response to all comments below. We sincerely thank the two reviewers for their useful and insightful suggestions in the first and second rounds of review and hope that the manuscript has been improved satisfactorily for publication in The Cryosphere.

Sincerely yours,

Kazuya Kusahara

**Response to Reviewer #1's comments**

(NB *italicized text in box* is comments from the reviewer. Some numbers with bold fonts were inserted for our convenience in addressing the comments.)

> *I found the manuscript strongly improved. I recommend this manuscript for publication. Below are very minor comments to improve readability and clarity.*
>
> **(1)** *Line 24: I would just say that the ACC is closer to the slope in all these regions, but the ACC does not reach the continental shelf, it simply brings warm water close to the shelf break.*
>
> **(2)** *Line 60-65: I would rephrase the text to say from the beginning that you are referring to the Shirase Glacier.*
>
> **(3)** *Figure 1: Include in the map where the Shirase Glacier Tongue and other glaciers are.*
>
> **(4)** *Line 216: Figure 3.*

Thank you very much for your careful reading and helpful suggestions/corrections.

**(1) (2)** We have rephrased the sentences.

**L23–24**

Both regions have a common oceanographic feature: southward deflection of the Antarctic Circumpolar Current brings CDW toward the continental shelves.

**L63–66**

In fact, looking at an estimate of basal melt rate from Rignot et al. (2013), a glacier tongue (Shirase Glacier Tongue, SGT) exhibits a high basal melt rate of $7\pm2$ m yr$^{-1}$, although the areal extent is small. The upstream glacier, Shirase Glacier in Lützow-Holm Bay (LHB), Enderby Land, East Antarctica, is one of the fastest laterally-flowing glaciers around Antarctica (Nakamura et al., 2010; Rignot, 2002).

**(3)** Thank you for your suggestion. We have added the abbreviations of the glaciers (SGT, KG, and SkG) in Fig. 1.

**(4)** We have corrected it (L217).

> **(5)** *Figure 3i: I would include a reference velocity (e.g. 1 m s-1) for the vectors in the figure.*
>
> **(6)** *Line 255-264: I would use the word "realistic" instead of "feasible".*
>
> **(7)** *Line 273: use acronym consistently in the subtitle (i.e. SGT).*
>
> **(8)** *Figure 8: Specify in the caption what the solid black line is (fast ice?).*
>
> **(9)** *Line 332-334 and Figure 10d: I don't fully understand here. The horizontal black line in the figure is the average difference in melt rate between the two simulations, or the SGT melt rate in a specific simulation? please clarify this both in the main text and in the figure caption.*
>
> **(10)** *Line 454: "Bringing warm water into the bay".*
>
> **(11)** *Line 923: a dot is missing at the end of the caption.*
>
> **(12)** *Line 467: "is more developed".*
>
> **(13)** *Line 550-563: In the Amundsen Sea models suggest that the undercurrent is present year-round. The seasonal appearance instead has been modelled in the Totten region. Please clarify this in the text.*

**(5)** Thank you for your suggestion. We have added alongshore unit (1 m/s) vector in the Fig. 3i.

**(6)(7)(10)(11)(12)** We have corrected them (L263, L274, L455, L928, L469).

**(8)** We have added a sentence in the caption (L860-861, "The thick zigzag black lines are the grounding line, the ice-front line, and the fast ice edge in the FI case.").

**(9)** We have rephrased the sentences in the text and caption.
**L333–336**
This is also confirmed in the difference of the SGT basal melt rate (Fig. 10d). The basal melt rate is rapidly decreased by approximately 4 m yr$^{-1}$ in the first winter, and then it continues to decline in a few years. Comparing with the melt rate difference in the last six years when the equilibriums are assumed to be reached in the FI and NOFI cases (black line and grey shade in Fig. 10d showing the mean and the standard deviations of the melt rate difference between the two cases), the melt rate deviation is in the standard deviation range after a few years.
**caption of Fig. 10 (L876–877)**
The black line and shade in panel (d) indicate the average of the SGT melt rate difference averaged over the period 2013–2018 between the FI and NOFI cases and the standard deviation.

**(13)** Thank you for your comment. We have simply removed the phrase ("in the summer season") for the correctness.
**L553-554**

[revised manuscript text omitted]